# Integrin-linked kinase controls retinal angiogenesis and is linked to Wnt signaling and exudative vitreoretinopathy

Hongryeol Park[1,13], Hiroyuki Yamamoto [1,8,13], Lucas Mohn[2,9], Lea Ambühl[2], Kenichi Kanai [1], Inga Schmidt[1], Kee-Pyo Kim [3], Alessia Fraccaroli[4,10], Silke Feil[2], Harald J. Junge [5,11], Eloi Montanez [4,12], Wolfgang Berger[2,6,7]* & Ralf H. Adams [1]*

Familial exudative vitreoretinopathy (FEVR) is a human disease characterized by defective retinal angiogenesis and associated complications that can result in vision loss. Defective Wnt/β-catenin signaling is an established cause of FEVR, whereas other molecular alterations contributing to the disease remain insufficiently understood. Here, we show that integrin-linked kinase (ILK), a mediator of cell-matrix interactions, is indispensable for retinal angiogenesis. Inactivation of the murine *Ilk* gene in postnatal endothelial cells results in sprouting defects, reduced endothelial proliferation and disruption of the blood-retina barrier, resembling phenotypes seen in established mouse models of FEVR. Retinal vascularization defects are phenocopied by inducible inactivation of the gene for α-parvin (*Parva*), an interactor of ILK. Screening genomic DNA samples from exudative vitreoretinopathy patients identifies three distinct mutations in human *ILK*, which compromise the function of the gene product in vitro. Together, our data suggest that defective cell-matrix interactions are linked to Wnt signaling and FEVR.

[1] Max Planck Institute for Molecular Biomedicine, Department of Tissue Morphogenesis, University of Münster, Faculty of Medicine, D-48149 Münster, Germany. [2] Institute of Medical Molecular Genetics, University of Zurich, CH-8952 Schlieren, Switzerland. [3] Max Planck Institute for Molecular Biomedicine, Department of Cell and Developmental Biology, D-48149 Münster, Germany. [4] Walter-Brendel-Centre of Experimental Medicine, University Hospital, LMU Munich, D-81377 Munich, Germany. [5] Department of Molecular, Cellular, and Developmental Biology, University of Colorado, Boulder, CO 80302, USA. [6] Zurich Center for Integrative Human Physiology, University of Zurich, Winterthurerstrasse 190, CH-8057 Zurich, Switzerland. [7] Neuroscience Center Zurich, University and ETH Zurich, Winterthurerstrasse 190, CH-8057 Zurich, Switzerland. [8]Present address: Department of Cardiovascular Medicine, Narita-Tomisato Tokushukai Hospital, 1-1-1 Hiyoshidai, Tomisato, Chiba 286-0201, Japan. [9]Present address: Department of Evolutionary Biology and Environmental Studies, University of Zurich, Winterthurerstrasse 190, CH-8057 Zurich, Switzerland. [10]Present address: Department of Medicine III, University Hospital, LMU Munich, Munich, Germany. [11]Present address: Department of Ophthalmology and Visual Neurosciences, University of Minnesota Medical School, Minneapolis, MN 55455, USA. [12]Present address: Department of Physiological Sciences, Faculty of Medicine and Health Sciences, University of Barcelona and Bellvitge Biomedical Research Institute (IDIBELL), 08907 Barcelona, Spain. [13]These authors contributed equally: Hongryeol Park, Hiroyuki Yamamoto. *email: berger@medmolgen.uzh.ch; ralf.adams@mpi-muenster.mpg.de

Appropriate growth of blood vessels is critical for the proper development of most organs and, consequently, defective angiogenesis is linked to several congenital human diseases. Familial exudative vitreoretinopathy or FEVR is a genetic disorder in which impaired angiogenesis in the retina of young children can lead to aberrant neovascularization and symptoms, such as the formation of retinal folds and tears, retinal detachment, or even total vision loss[1]. Among the genes associated with FEVR, components of the Wnt/β-catenin signaling pathway are highly prevalent. These are the genes encoding the growth factor Norrin (*NDP*), the receptor Frizzled-4 (*FZD4*), the low-density lipoprotein receptor-related protein 5 (*LRP5*), and tetraspanin-12 (*TSPAN12*), which forms transmembrane receptor complexes with LRP5 and Frizzled-4[2,3]. For other FEVR-associated mutations, which affect a zinc finger family DNA-binding protein (*ZNF408*), a kinesin family molecular motor protein (*KIF11*) and a basic helix–loop–helix family transcription factor (*ATOH7*), it is less clear whether they are functionally related to Wnt/β-catenin signaling[4–7]. While several of the genes linked to FEVR directly regulate endothelial cell (EC) behavior in various animal models[2,3,6,8–11], ATOH7 (also known as Math5) is an important regulator of neuronal and glial development in the retina[12,13]. Thus, some vitreoretinopathies might be caused by primary defects outside the vasculature.

Integrin family receptors and associated components of the cell–matrix adhesion machinery, such as focal adhesion kinase, integrin-linked kinase, kindlin, or talin are also recognized key regulators of EC function and angiogenesis[14–17]. ILK is an important mediator of integrin signaling and forms a tripartite complex (termed IPP) with PINCH and parvin adapter proteins, which together regulate cell contractility, actin, and microtubule organization, and cytoskeletal dynamics[18]. Constitutive, EC-specific knockout of *Ilk* in mice leads to embryonic lethality due to impaired vascularization of the embryo proper and the labyrinthine layer in the placenta[19]. In ECs derived from *Ilk* knockout embryonic stem cells, the organization of microtubules and cortical actin filaments, the positioning of the scaffolding protein caveolin 1, and the cellular responses to vascular endothelial growth factor (VEGF) and epidermal growth factor (EGF) are compromised[20]. Knockdown of *ILK* expression also impairs the migration of human umbilical vein ECs (HUVECs) toward VEGF and pharmacological ILK inhibition leads to reduced tumor growth and angiogenesis in a xenograft model[21]. However, the function of ILK in postnatal physiological angiogenesis in vivo has not been studied so far and remains little understood.

In the current study, we use inducible EC-specific approaches to show that the murine *Ilk* gene is indispensable for the vascularization of the postnatal retina, a well-established model system of angiogenesis[22,23]. Vascular defects in *Ilk* mutants resemble phenotypes seen after inducible inactivation of the gene for α-parvin (*Parva*) and those in previously established mouse models of FEVR[24]. Furthermore, we identify three loss-of-function mutations in human *ILK* in genomic DNA samples from exudative retinopathy patients, which links defective cell–matrix interactions to the development of this disease.

## Results

### Defects after inducible, EC-specific inactivation of *Ilk*. To circumvent the embryonic lethality of EC-specific *Ilk* mutants[19] and uncover the function of ILK in postnatal retinal angiogenesis, mice carrying a loxP-flanked allele of the gene[25] were bred to tamoxifen-inducible *Pdgfb-iCre*[23] transgenic animals. Tamoxifen was initially administered at postnatal day (P1) to P3 followed by analysis of the resulting *Pdgfb-iCre*[+/−] *Ilk*[lox/lox] (*Ilk*[iECKO]) mutant and Cre-negative control littermate retinas at P6 (Fig. 1a), which

allows detailed analysis of the growing superficial vascular plexus. Immunostaining of retina sections confirmed that this approach abolished ILK expression specifically in ECs but not in *Ilk*[iECKO] perivascular cells (Supplementary Fig. 1A). Vessel outgrowth as well as vessel density and branching are reduced in *Ilk*[iECKO] retinas relative to littermate controls (Fig. 1b, e). Filopodia-extending endothelial sprouts at the edge of the growing mutant vessel plexus are present and even more abundant than in P6 control samples but remain short and display an abnormally thick morphology with an enlarged lumen, as indicated by immunostaining of ICAM2, a cell adhesion molecule at the apical EC surface facing the vessel lumen (Fig. 1b, c, e). Immunostaining of transmembrane proteins connecting adjacent ECs, namely the adherens junction protein VE–cadherin and the tight junction protein Claudin 5, uncovers a highly increased complexity of EC–EC contacts at the distal *Ilk*[iECKO] vessel plexus (Fig. 1d; Supplementary Fig 1B, C). Immunostaining for the transcription factor Erg, which decorates endothelial nuclei, shows dense clustering of mutant ECs within the thick *Ilk*[iECKO] sprouts (Fig. 1d). This clustering of ECs at the angiogenic front and in the short sprouts together with the reduced outgrowth of the vessel plexus is consistent with the important role of integrins and ILK as regulators of cell migration[26–28]. In line with the overall reduced vascular coverage and reduction of vessel branch points in mutant retinas (Fig. 1e), *Ilk*[iECKO] ECs show strongly reduced proliferation relative to control samples (Fig. 1f, g).

After the initial outgrowth of the superficial vessel plexus, retinal vessels sprout perpendicularly into the deeper retina and successively form two further capillary layers[22]. This process was analyzed in P14 mice in which *Ilk* inactivation in ECs had been induced by three tamoxifen injections at P3, P5, and P7 (Fig. 2a). While three vessel layers are clearly distinguishable in P14 control samples, vascularization of the deeper retina is defective in *Ilk*[iECKO] mutants (Fig. 2b–e). Detailed analysis of the *Ilk*[iECKO] retinal vasculature reveals that the perpendicular sprouting of vessels from the superficial plexus is strongly compromised. Mutant sprouts are stalled in the vicinity of the nerve fiber layer and form thick protrusions (Fig. 2c, d). Normal vessel remodeling and maturation in the superficial retina involves the pruning of side branches and a gradual reduction of capillary diameters[29]. These remodeling processes are impaired in P14 *Ilk*[iECKO] retinas, which display enlarged vessel diameters, excess branch points and increased endothelial coverage in the superficial vessel plexus (Fig. 2b, c). Thus, loss of endothelial *Ilk* function not only leads to defective endothelial sprouting but also interferes with vessel remodeling.

### FEVR-like vascular defects in *Ilk* mutant mice. Defective vascularization of the deeper retina has been reported for mutant mice lacking critical regulators of actin dynamics[30–32] as well as numerous components of the Wnt signaling pathway[2]. Mutations in several human Wnt pathway genes have been also linked to FEVR and, in particular, incomplete vascularization of the peripheral retina combined with increased vessel leakage and hemorrhaging in the eye[10,11,33,34]. Many of these disease features are reproduced by EC-specific mouse mutants for *Fzd4* and other genes[2,3], and are also phenocopied by *Ilk* mutant mice. P14 *Ilk*-[iECKO] mutants are reduced in size and body weight (Fig. 3a) and development of the retinal vasculature is severely compromised (Fig. 3b). Endothelial sprouts penetrating from the vitreal surface have been reported to terminate in thick, roundish clusters in EC-specific *Fzd4* mutants and global *Fzd4*, *Lrp5*, or *Ndp* knockout mice[2,3]. Similar clusters are abundant in the *Ilk*[iECKO] retinal vasculature (Fig. 3d and Supplementary Fig. 2A). These endothelial clusters are associated with macrophages, which are also

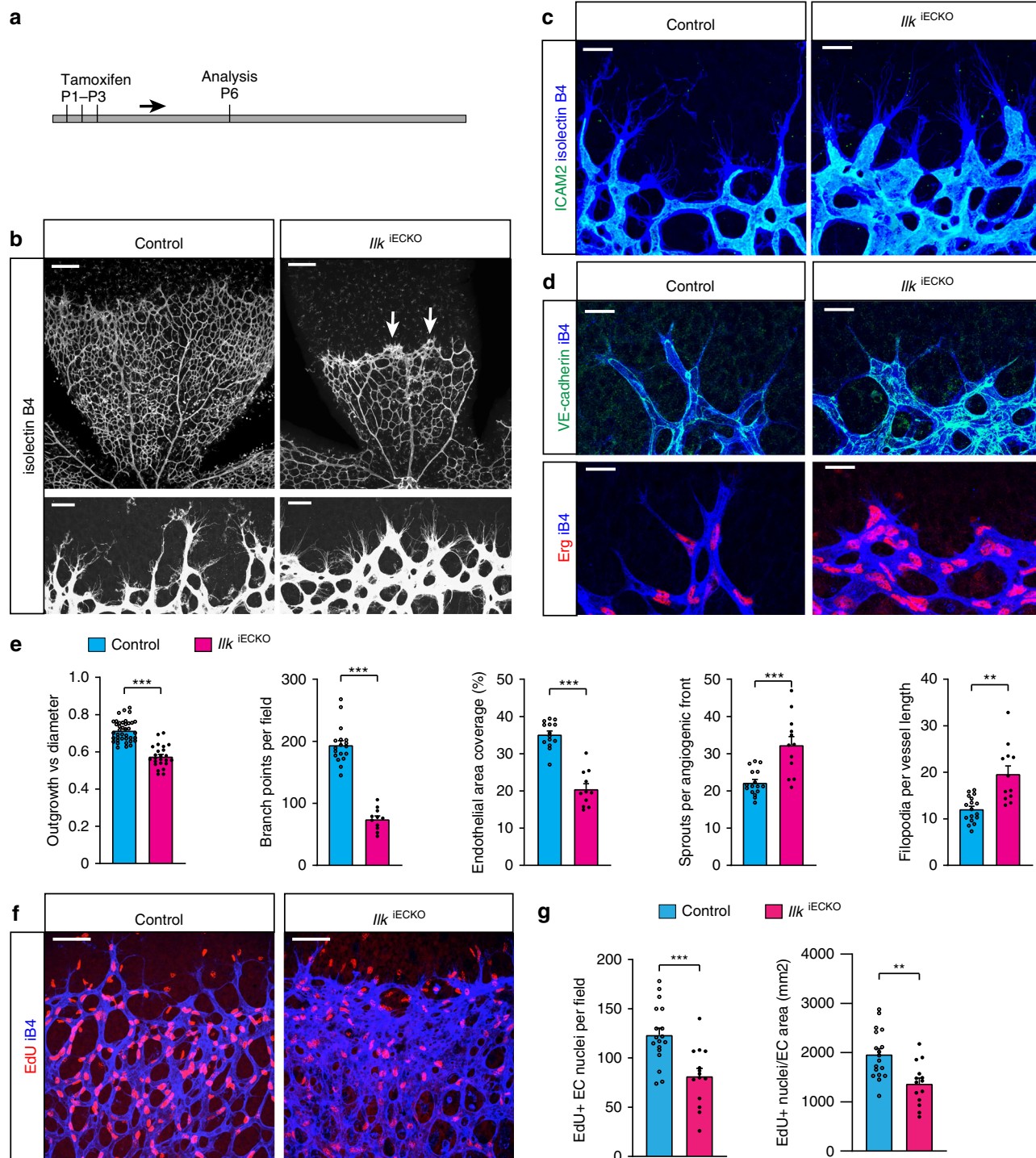

**Fig. 1** Postnatal EC-specific *Ilk* inactivation induces sprouting defects in retina. **a** *Ilk*iECKO mutants and littermate controls received intraperitoneal tamoxifen injections once daily from P1–3 and were analyzed at P6. **b** Low-magnification overview (top panels) and high-magnification images of angiogenic growth front (bottom panels) in isolectin B4 stained control and *Ilk*iECKO P6 retinas, respectively. Note reduction of vessel density and radial outgrowth in *Ilk*iECKO mutants despite the abundance of filopodia-extending ECs. Abnormally thick sprouts and EC clusters are detected in *Ilk*iECKO samples (arrows). Scale bar, 200 μm (top panels), 25 μm (bottom). **c** ICAM2 (green) and isolectin B4 (blue) staining in P6 control and *Ilk*iECKO retinas. The lumen of vessels, indicated by apical ICAM2 signal, is enlarged in the leading edge of *Ilk*iECKO capillaries. Scale bar, 25 μm. **d** Confocal images showing VE-cadherin (green) and Erg (red) immunostaining in combination with isolectin B4 (blue) in control and *Ilk*iECKO retinas, as indicated. Loss of *Ilk* leads to clustering of ECs at the growth front and increases the complexity of EC–EC junctions. Scale bar, 25 μm. **e** Quantification of outgrowth vs. retina diameter, branch points per field and EC area coverage in overview images of control and *Ilk*iECKO retinas. Sprouts per angiogenic front and filopodia per vessel length at the front area are also quantified. Error bars, s.e.m. *p* values (***p* < 0.001, ***p* < 0.01), Student's *t* test (*n* = 8 control and 6 mutants retinas). **f, g** Incorporation of EdU in isolectin B4-stained control and *Ilk*iECKO retinas. Representative confocal images (**f**) and quantification (**g**) of proliferating ECs both in number per field and relative to EC-covered area. Scale bar, 50 μm. Error bars, s.e.m. *p* values (***p* < 0.001, ***p* < 0.01), Student's *t* test (*n* = 10 control and 8 mutant retinas). Source data are provided as a Source Data file

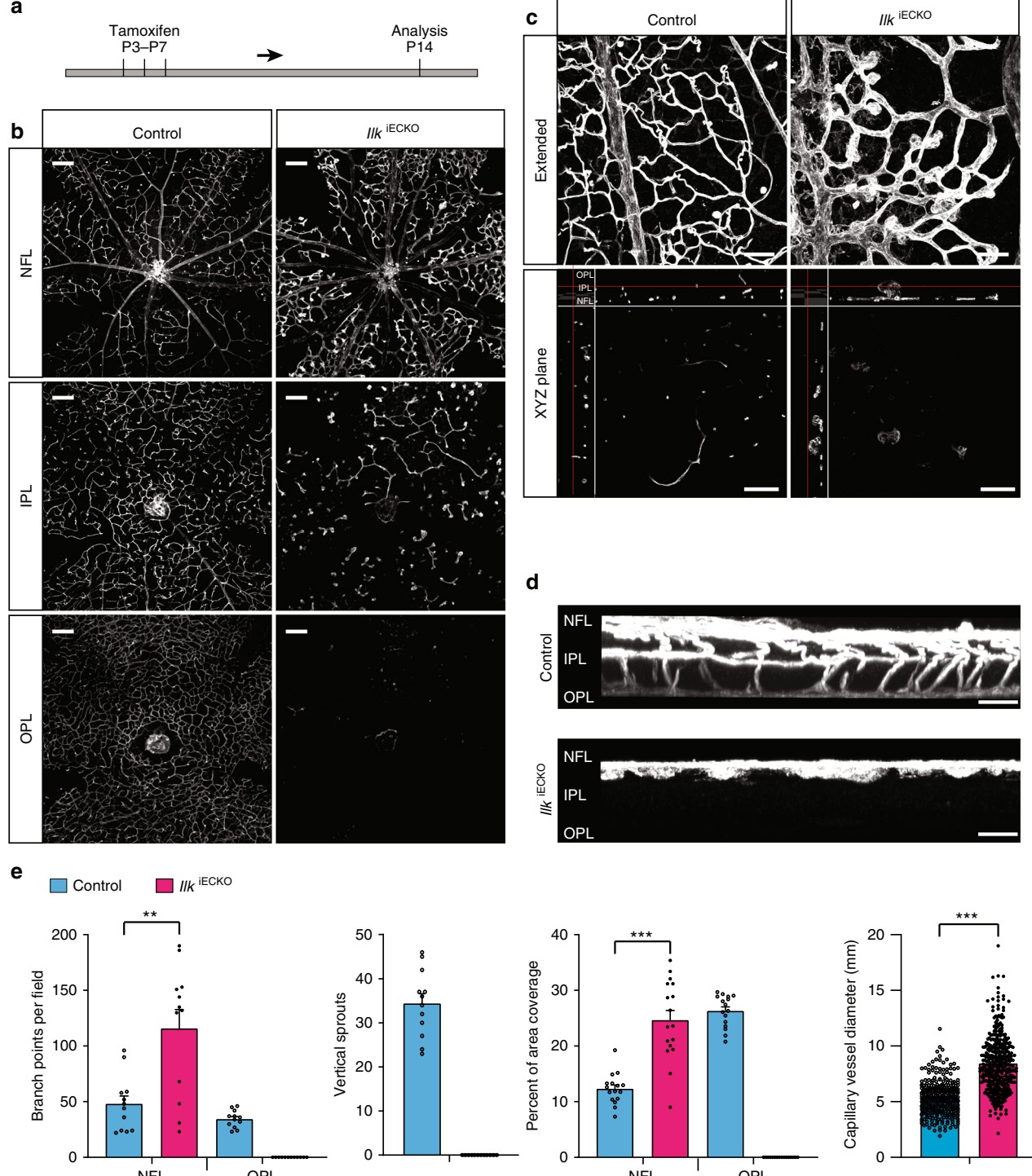

**Fig. 2** ILK is indispensable for plexus formation in the deeper retina. **a** *Ilk*iECKO mutants and littermate controls received intraperitoneal tamoxifen injections every other day from P3–7 and were analyzed at P14. **b** Isolectin B4-stained *Ilk*iECKO mutants and littermate control retinas. Optical sections of z-stacked confocal images were divided to represent the nerve fiber layer (NFL), inner plexiform layer (IPL), and outer plexiform layer (OPL). Note strong reduction of the *Ilk*iECKO IPL and OPL vasculature. Scale bar, 200 μm. **c**, **d**, XY-extended, ZY-extended, and XYZ plane view high magnification images of control and *Ilk*iECKO retinas stained with isolectin B4. Vertical sprouting is impaired and abnormal, bulging endothelial structures decorate dilated vessels in *Ilk*iECKO retinas. Scale bar, 100 μm (top) and 50 μm (bottom) in (**c**), 25 μm in (**d**). **e** Quantification of branch points, vascular area coverage, vertical sprouts and capillary vessel diameter, as shown in (**b**) and (**c**). Error bars, s.e.m. *p* values (***$p < 0.001$, **$p < 0.01$), Student's *t* test ($n = 6$ retinas/group). Source data are provided as a Source Data file

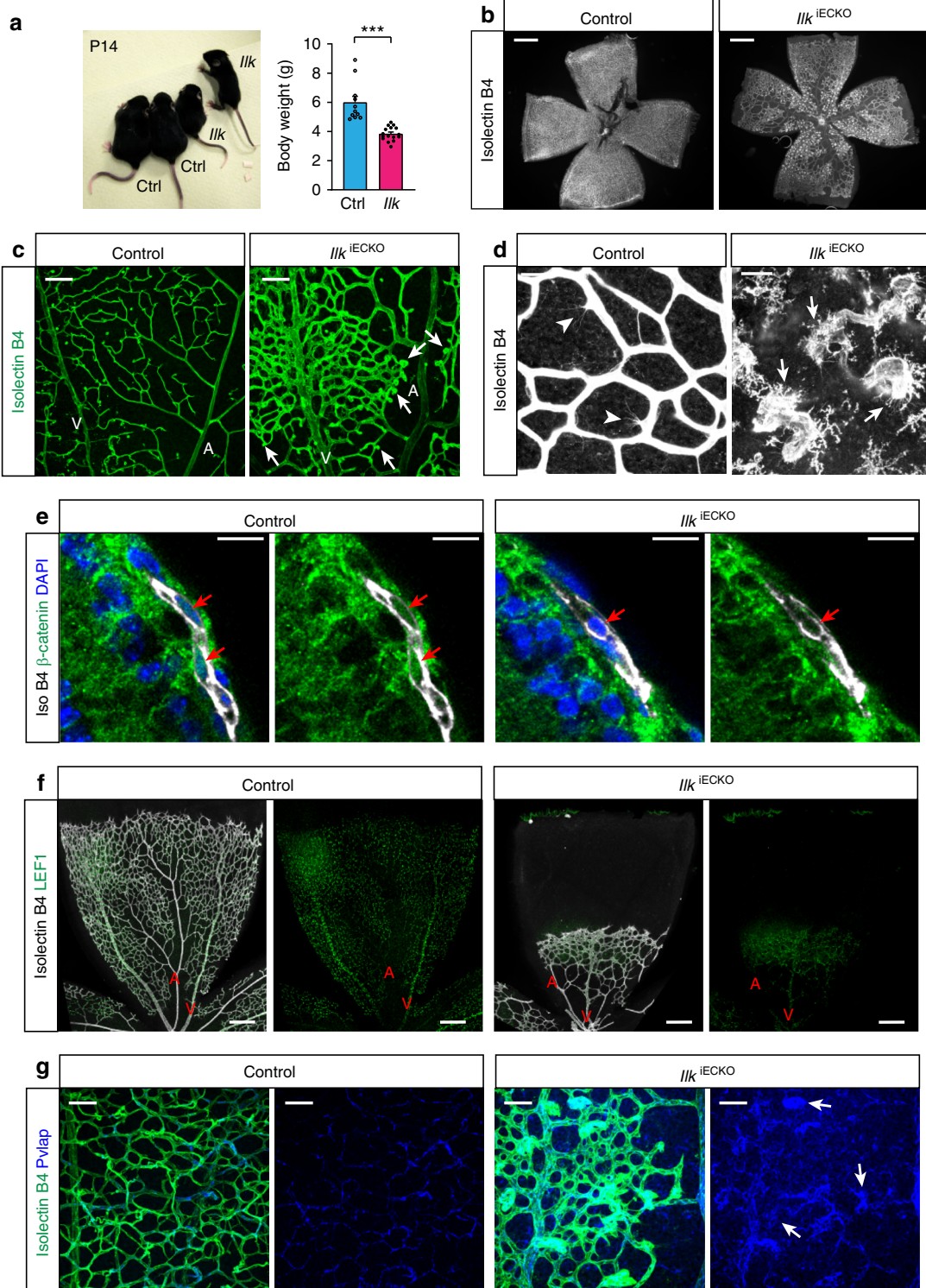

**Fig. 3** Vascular abnormalities in *Ilk* mutants phenocopy aspects of FEVR. **a** Picture and body weight analysis of P14 *Ilk*[iECKO] mutants and littermate controls. Error bars, s.e.m. *p* values (***$p < 0.001$), Student's *t* test ($n = 6$ animals/group). **b** Overview images of the isolectin B4-stained P14 control and *Ilk*[iECKO] retinal vasculature. Scale bar, 500 μm. **c** Confocal images of the isolectin B4-stained vasculature in the control and *Ilk*[iECKO] nerve fiber layer. Arteries (A), veins (V), and bulging EC clusters emerging from mutant capillaries (arrows) are indicated. Scale bar, 100 μm. **d** High contrast images of isolectin B4-stained vessels in the P14 nerve fiber layer. Filopodia-extending sprouts (arrowheads) can be seen on control capillaries, whereas blunt ends of *Ilk*[iECKO] vessels are associated with isolectin B4+ macrophages (arrows). Scale bar, 25 μm. **e** Confocal images of β-catenin (green) and isolectin B4 (white) stained control and *Ilk*[iECKO] P6 retina sections. Presence (control) or absence (*Ilk*[iECKO]) of β-catenin in endothelial nuclei (red arrows) in images with or without DAPI signal. Scale bar, 20 μm. **f** Overview images of LEF1 and isolectin B4 stained retina of control and *Ilk*[iECKO]. Arteries (A) and veins (V) are indicated. Scale bar, 200 μm. **g** Confocal high magnification images of Pvlap (blue) and isolectin B4 (green) stained control and *Ilk*[iECKO] retinas. Note enhanced Pvlap staining of *Ilk*[iECKO] vessels and strong signal in distal EC clusters (arrows). Scale bar, 50 μm. Source data are provided as a Source Data file

stained by isolectin B4, and the number of F4/80-positive macrophages is increased in $Ilk^{iECKO}$ retinas (Fig. 3d and Supplementary Fig. 2B).

Arguing that the similarities between the $Ilk^{iECKO}$ phenotype and defects in the Wnt/β-catenin mutant retinal vasculature might reflect a functional link, β-catenin immunosignals are readily detectable in control but not in $Ilk^{iECKO}$ EC nuclei (Fig. 3e). The transcription factor LEF1 (lymphoid enhancer-binding factor 1), one of the transcriptional mediators of Wnt/β-catenin signaling, marks nuclei in venous and capillary ECs in P6 and P14 control retinas (Fig. 3f; Supplementary Fig. 2C). Intriguingly, venous LEF1 expression is strongly reduced in P14 $Ilk^{iECKO}$ retinas, whereas arteries show ectopic LEF1 immunosignal (Supplementary Fig. 2C). Further arguing for defective maturation and integrity of the $Ilk^{iECKO}$ vasculature, pronounced hemorrhaging is seen in mutant retinas and brain whole-mounts and sections (Supplementary Fig. 3A, B). The latter is most prominent in the cerebellum, which is similar to brain hemorrhaging reported for Fzd4-deficient mice[2,3] and consistent with an important role of Wnt signaling in the establishment of the blood–brain barrier (BBB) and blood–retina barrier (BRB)[35,36]. Expression of Pvlap/PV1, which forms diaphragms in endothelial fenestrae, is characteristic for immature, highly permeable or fenestrated vessels, but is downregulated in the normal postnatal vasculature of the central nervous system (CNS)[37–39]. While only weak Pvlap immunostaining is detectable in P14 control capillaries, signals are strongly elevated in $Ilk^{iECKO}$ vessels and, in particular, in roundish endothelial clusters (Fig. 3g). These findings establish that loss of endothelial $Ilk$ compromises BBB and BRB function, leads to hemorrhaging and causes FEVR-like vascular defects in mice.

ILK forms functional, heterotrimeric IPP complexes with the adapter proteins PINCH (particularly interesting Cys-His-rich protein) and parvin, which stabilize each other, link integrin receptors to the actin cytoskeleton and control downstream signal transduction[18,40]. Endothelial α-parvin (encoded by the gene $Parva$) is essential for angiogenesis in embryonic and postnatal mice, which involves the regulation of the cell adhesion molecule VE-cadherin[24]. Inactivation of $Parva$ in the postnatal endothelium leads to the appearance of EC clusters and sprouting defects resembling those seen in $Ilk^{iECKO}$ retinas (Fig. 4a–c). Likewise, vascularization of the deeper retina is strongly impaired in EC-specific $Parva^{iECKO}$ mutants (Fig. 4d). These results support that physiological retinal angiogenesis requires ILK, α-parvin and therefore presumably also the formation of IPP complexes.

As global loss of $Ilk$ gene function in mice is incompatible with preimplantation development, we also investigated the vasculature of heterozygous, EC-specific mutants ($Ilk^{iECKO/+}$). While these mice appear normal and display extensive vascularization of both the superficial and deeper retina (Fig. 5a), the number of vessel projections from the vitreal surface as well as vessel density and number of branch points in the deep retinal plexus are significantly reduced (Fig. 5a–d). Despite of the essential roles of $Ilk$ in preimplantation development, it has been reported that heterozygous knockouts appear normal without any overt anatomical or behavioral abnormalities[41]. To address potential vascular alterations in global mutants lacking one functional allele of $Ilk$, we interbred $Ilk^{lox/lox}$ conditional animals with the PGK-Cre deleter strain[42]. Constitutive $Ilk^{+/−}$ heterozygous knockout mice generated with this breeding strategy and control littermates show no overt differences in body weight or the vascularization of the superficial retina (Supplementary Fig. 4A, B). However, vascularization of the deeper retina is compromised, as indicated by significant reductions in the number of vessel branch points, vascular density, and vertical vessel projections from the vitreal

surface in P14 $Ilk^{+/−}$ retinas relative to littermate controls (Supplementary Fig. 4C, D). These results establish that EC-specific $Ilk^{iECKO/+}$ and global $Ilk^{+/−}$ mutants develop similar vascular phenotypes and highlight that ECs respond sensitively to the loss of a single $Ilk$ allele.

**ILK variants in human patients**. Based on the identification of FEVR-like defects in $Ilk^{iECKO}$ retinas, a cohort of genomic DNA samples from 208 exudative vitreoretinopathy patients, most of them sporadic cases with unknown genetic cause, were screened by Sanger sequencing for mutations in the 12 protein coding exons as well as flanking splice sites of the human $ILK$ gene. This approach revealed three distinct missense mutations in the open reading frame of $ILK$ in four patients but in none of the control samples (420 controls = 840 alleles). Two of the patients are heterozygous for the variant p.Arg211Cys (NM_004517.2:c.631C > T), which converts an arginine into a cysteine residue within the phosphatidylinositol 3,4,5-triphosphate (PIP3)-binding region of ILK (Fig. 6a–c). Arg211 is highly conserved among vertebrate and invertebrate species and it was previously shown that an ILK Arg211Ala variant protein is impaired in its ability to induce protein kinase B (PKB/Akt) phosphorylation in cultured cancer cell lines[43,44]. A second Ilk variant, p.Leu53Met (NM_004517.2:c.157T > A) alters a highly conserved leucine residue within the N-terminal ankyrin repeat region, which is important for binding of PINCH and ILK associated serine/threonine phosphatase (ILKAP)[45] (Fig. 6a–c). The variant p.Arg317Gln (NM_004517.2:c.950G > A) alters a residue within the ILK kinase domain (Fig. 6a–c)[45]. The catalytic activity of ILK was previously shown to be dispensable for normal development and tissue function in knock-in mice, but residues with the kinase domain are nonetheless critical for α-parvin binding and thereby for ILK function[46–48]. Thus, the three newly identified $ILK$ variants are likely to affect the biological activity of the gene product. Moreover, all three variants occur more frequently than mutations in human $NDP$ or other FEVR-associated genes, but are, nevertheless, not very frequent (p.Leu53Met: 0.035%; pArg211Cys: 0.084%; p.Arg317Gln: 0.0016%) and mostly confined to European subjects within the data set of the Exome Aggregation Consortium (ExAC)[49]. Consistent with the essential roles of ILK in development, no homozygotes for the three alleles were found in the ExAC database.

**ILK variants represent partial loss-of-function alleles**. To study the biological activity of the three variants identified above, we designed human $ILK$ constructs, which are resistant to siRNA targeting without changing the amino acid sequence of the resulting gene products (Fig. 6d). The detection of these proteins was facilitated by an aminoterminal fusion to green fluorescent protein (GFP-ILK) in some experiments. For knockdown of endogenous $ILK$ expression in HUVECs, we used a siRNA targeting exon 2, which strongly reduces the expression of endogenous ILK (Fig. 7a). Knockdown of $ILK$ expression impairs cell spreading and leads to a pronounced reduction of paxillin-containing focal adhesions and focal complexes in transfected HUVECs (Fig. 7b). These defects are rescued by expression of siRNA-resistant GFP-ILK but not by GFP alone. Remarkably, all three mutated versions of GFP-ILK fail to restore normal HUVEC spreading and focal adhesion organization (Fig. 7b), which is most obvious for the L53M and R211C variants.

Interactions between ILK, PINCH and parvin are critical for the stability of these proteins in the tripartite IPP complex[40]. Immunoprecipitation of overexpressed and GFP-tagged wild-type ILK leads to pull-down of associated PINCH and α-parvin, whereas PINCH co-immunoprecipitation is strongly reduced for

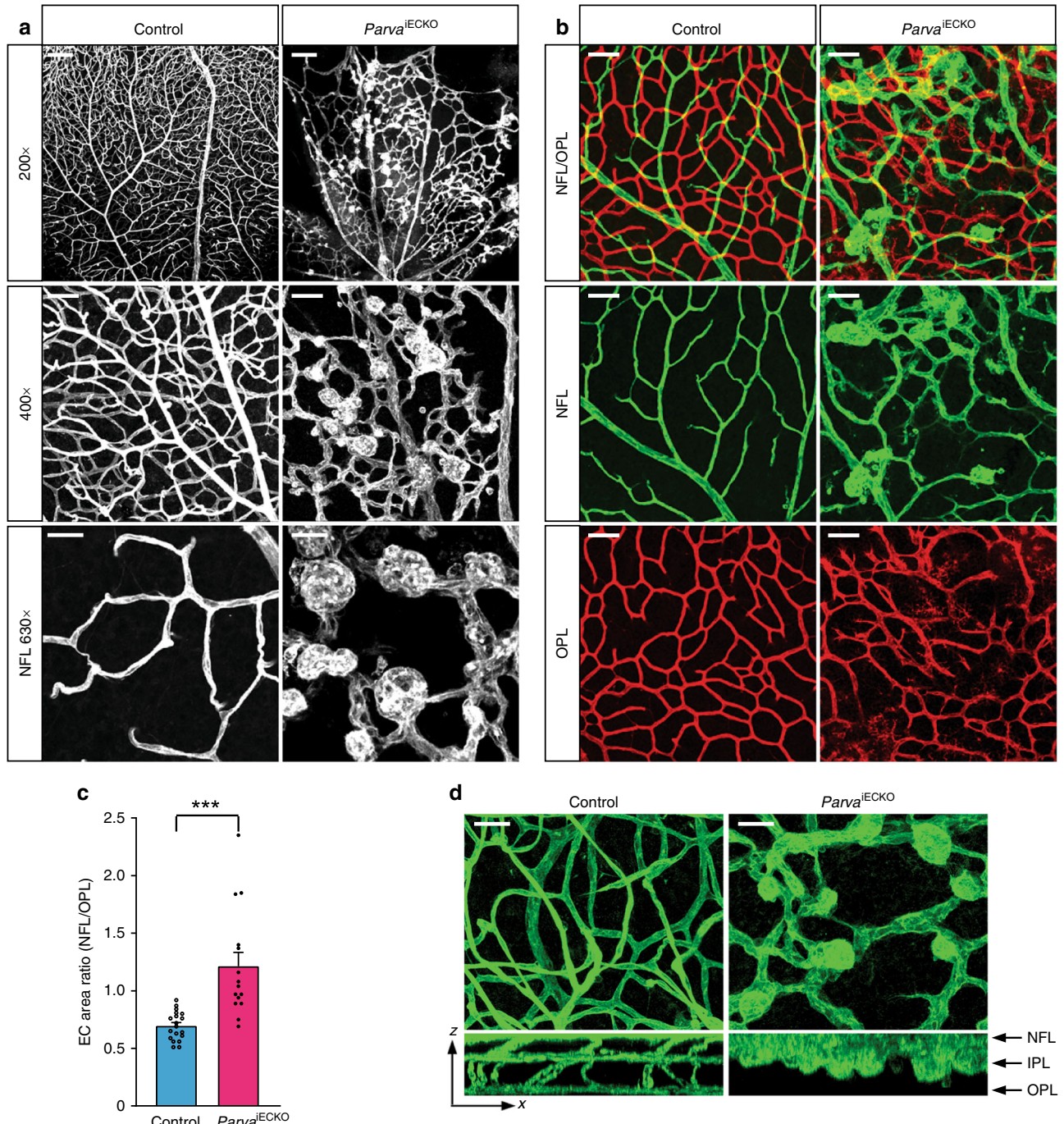

**Fig. 4** EC-specific knockout of *Parva* phenocopies *Ilk*iECKO vascular defects. **a** Confocal images of isolectin B4 stained P16 control and *Parva*iECKO retinas. Note large and blind-ended EC bulges in the *Parva*iECKO but not the control vasculature. Scale bar, 150 μm (top), 50 μm (center), and 25 μm (bottom). **b**, **c** Images of isolectin B4-stained retinal vessels with pseudo-coloring of the NFL (green) and OPL (red) vasculature (**b**). *Parva*iECKO EC bulges are confined to the NFL, whereas the mutant OPL vessel density is decreased resulting in a significant increase of the NFL/OPL EC area ratio (**c**). Scale bar, 50 μm in (**b**). Error bars, s. e.m. *p* values (****p* < 0.001), Student's *t* test (*n* = 5 control and 4 mutant retinas). **d** XZ projection images of isolectin B4-stained control and *Parva* iECKO retinas. Mutant NFL vessels are enlarged and vertical sprouting is impaired. Scale bar, 25 μm. Source data are provided as a Source Data file

L53M, R211C, and R317Q ILK (Fig. 7c). In contrast, expression levels and ILK immunoprecipitation efficiencies are indistinguishable for wild-type GFP-ILK and all three variants (Fig. 7c). This is even more obvious after siRNA-mediated knockdown of *ILK* expression, which strongly reduces endogenous ILK protein levels but also the stability of PINCH and α-parvin in HUVECs (Supplementary Fig. 5B). These defects are rescued by re-expression of wild-type GFP-ILK but not by the L53M and R317Q variants. R211C has an intermediate effect in this assay

and allows the pull-down of associated PINCH and α-parvin, which is strongly reduced for L53M and R317Q (Supplementary Fig 5B).

Previous work has linked ILK to phosphorylation of PKB/AKT in cultured cancer cells and fibroblasts[43,50], which also reflects ILK-dependent cell–matrix interactions and integrin activation. AKT phosphorylation (in Ser473) but not total AKT is reduced after *ILK* knockdown in HUVECs, which is not rescued by expression of the R211C, R317Q, or L53M ILK variant (Fig. 7a;

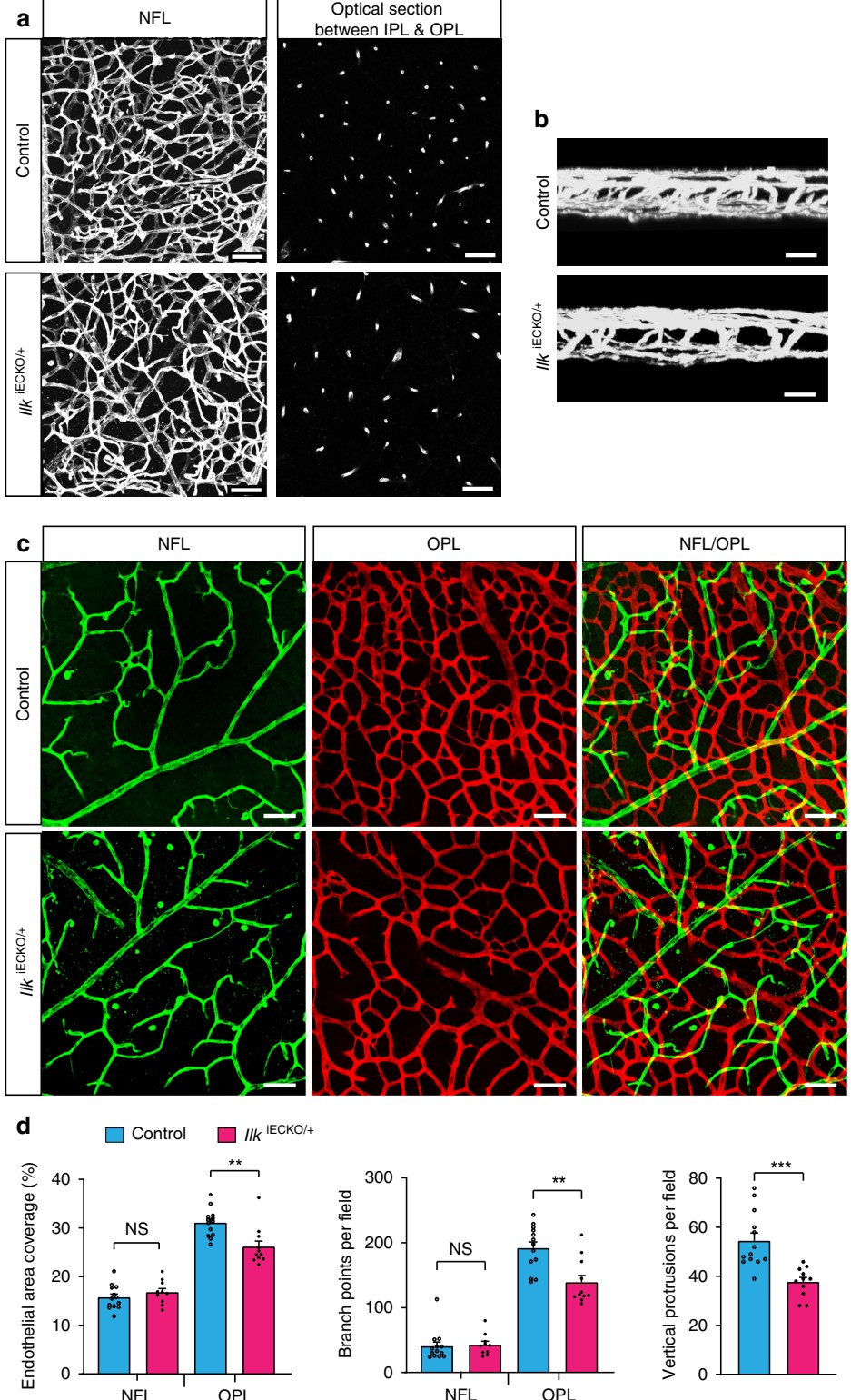

**Fig. 5** Vascular defects in *Ilk*iECKO/+ heterozygotes. **a**, **b** Extended focus (**a**:XY; **b**:XZ) and optical section images (right panels in **a**) of isolectin B4-stained P17 control and *Ilk*iECKO/+ retinal wholemounts. Vessel density and vertical sprouts between IPL and OPL are reduced in *Ilk*iECKO/+. Scale bar, 50 µm in (**a**) and 25 µm in (**b**). **c**, **d** Images of isolectin B4-stained control and *Ilk*iECKO/+ retinal vessels with pseudo-coloring of the NFL (green) and OPL (red) vasculature (**c**). Scale bar, 50 µm. *Ilk*iECKO/+ OPL EC area, branch points and vertical sprouts are reduced, whereas NFL vessels are comparable to control samples (**d**). Error bars, s.e.m. *p* values (\*\*\**p* < 0.001, \*\**p* < 0.01, ns not significant), Student's *t* test (*n* = 3 retinas/group). Source data are provided as a Source Data file

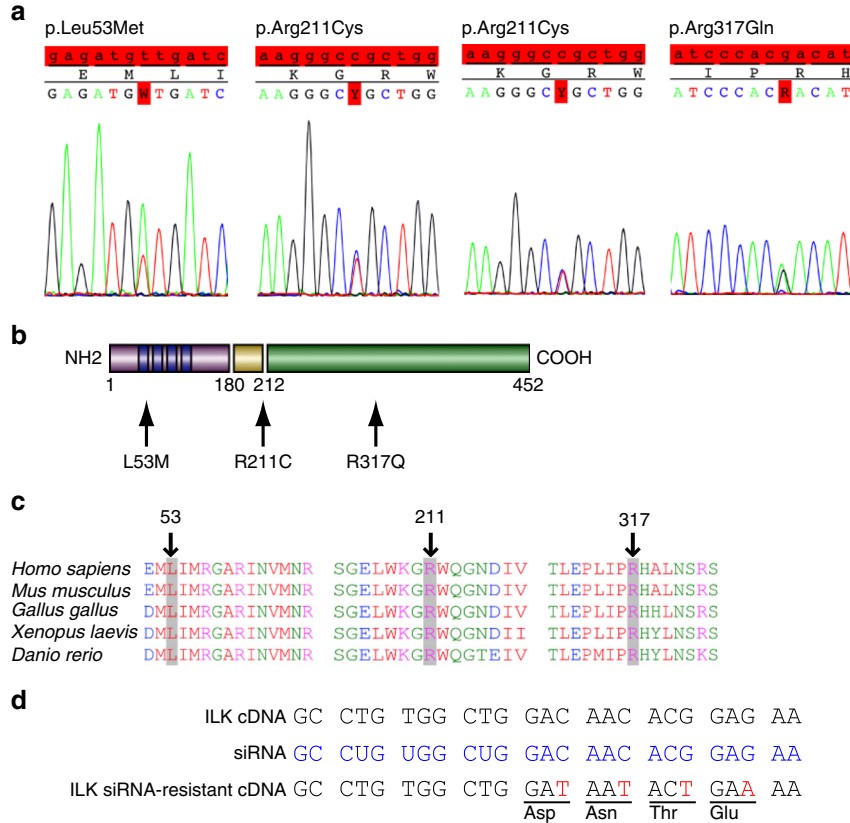

**Fig. 6** *ILK* point mutations in human EVR patients. **a**, **b** Sanger sequencing of three different *ILK* point mutations identified in four patients (**a**). The corresponding changes in the primary sequence and location of the affected amino acid residues (red box) within different domains of the gene product are indicated (**b**). Mutation NM_004517.2:c.157T > A changes Leucine to Methionine at position 53 (p.Leu53M) of the ILK protein, NM_004517.2:c.631C > T changes Arginine to Cysteine at position 211 (p.Arg211Cys), and NM_004517.2:c.950G > A changes Arginine to Glutamine at position 317 (p.Arg317Gln). **c** Alignment of ILK amino acid sequences in different species, as indicated. Altered amino acid residues in FEVR patients are highlighted in gray and marked by arrows. All mutated sites are highly conserved. **d** *ILK* siRNA targeting strategy and generation of siRNA-resistant cDNA sequences, which avoid siRNA binding without changing the primary sequence of the gene product

Supplementary Fig 5A). Moreover, endothelial phospho-AKT immunostaining is strongly diminished in *Ilk*[iECKO] retina sections relative to littermate control (Fig. 7d). *ILK* knockdown in HUVECs reduces the levels of β-catenin, which is rescued by re-expression of wild-type ILK, but not any of three variant proteins (Fig. 7a; Supplementary Fig 5A). Arguing further for a role of ILK in Wnt signaling, knockdown of integrin-linked kinase in HUVECs or HRMVECs (primary human retinal microvascular ECs) impairs the upregulation of the Wnt target genes *AXIN2*, *SOX17*, and *FLT1* in response to stimulation with recombinant hNorrin (Fig. 7e and Supplementary Fig. 5C). hNorrin-induced upregulation of *AXIN2* in this setting is significantly restored by re-expression of wild-type ILK but not any of the three variants (Fig. 7e).

ILK has oncogenic activity[44,51], which can be readily assessed by focus formation assays in culture. Overexpression of wild-type ILK in NIH 3T3 cells induces the formation of foci significantly above the level seen for vector control (Supplementary Fig. 5D). In contrast, the number of foci obtained after overexpression of any of the three ILK variants is comparable to vector control (Supplementary Fig. 5D). Together, all these assays indicate that the sequence changes in L53M, R211C, and R317Q ILK reduce the biological activity of the gene product, indicating that the variants identified in FEVR patients are functional hypomorphs. Our data also show that ECs react sensitively to changes in ILK level and activity, which involves changes in cell morphology, behavior, and Wnt signaling.

**Discussion**

ILK is an indispensable regulator of cell–matrix signaling in many different cell types and organ systems, and, accordingly, global inactivation of the murine *Ilk* gene leads to impaired epiblast polarization and preimplantation lethality[41]. Likewise, constitutive loss of *Ilk* in ECs leads to embryonic lethality at mid-gestation[19] as soon as development requires the proper function of the vasculature and placenta. The current study has used EC-specific and inducible genetic approaches, which have provided insights into *Ilk* function in postnatal angiogenesis and have linked the gene to exudative vitreoretinopathy. The vascular defects of *Ilk*[iECKO] mutants are phenocopied by EC-specific inactivation of *Parva*, which highlights the importance of the interactions between ILK, PINCH, and parvin proteins in the IPP complex. Our data also indicate that β-parvin cannot compensate for the loss of α-parvin in ECs and is consistent with previous reports assigning distinct functional roles to the two proteins in mammalian cells[52]. The IPP complex associates with integrin heterodimers and thereby couples extracellular signals to a variety of intracellular processes such as signal transduction, cytoskeletal dynamics, or membrane trafficking[18]. Postnatal, EC-specific mutant mice lacking integrin β1 (*Itgb1*[iECKO]), a subunit that can form complexes with 12 distinct integrin α chains, also develop a range of vascular defects such as hemorrhaging and compromised angiogenesis[53]. However, *Itgb1*[iECKO] mutants display EC hyperproliferation in retinal veins and at the angiogenic growth front, whereas EC proliferation is reduced in *Ilk* and

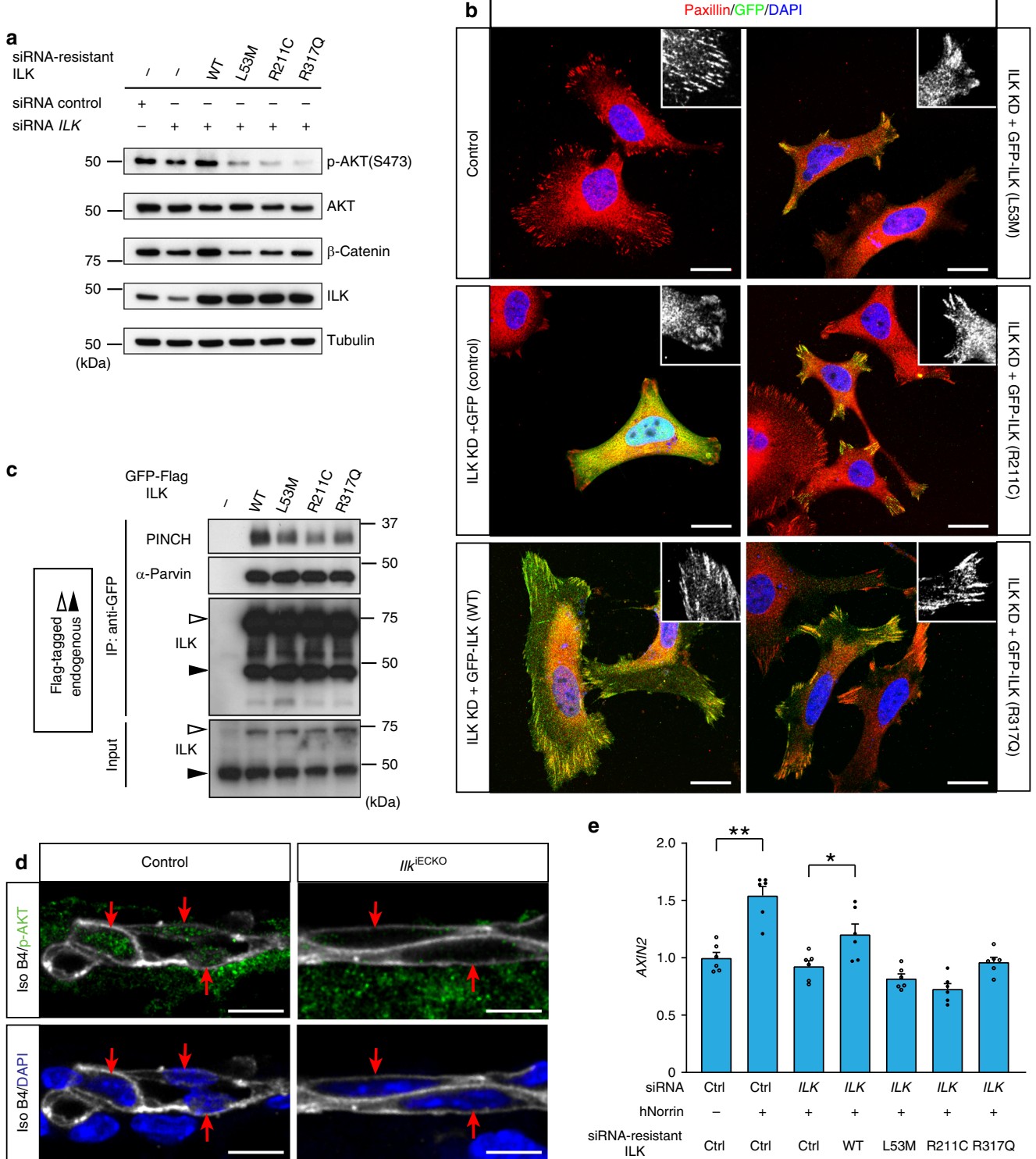

**Fig. 7** Functional consequences of ILK mutant alleles. **a** Western blot analysis of β-catenin and AKT Ser473 phosphorylation in HUVEC after siRNA-mediated knockdown of endogenous *ILK* and expression of siRNA-resistant wild-type (WT) ILK and variant versions, as indicated. Quantitation provided in Supplementary Fig. 5A. **b** Paxillin (red) and DAPI (nuclei, blue) stained HUVECs showing the changes in cell morphology and focal adhesion formation (high-magnification insets) after knockdown (KD) of endogenous *ILK* expression and transfection with either GFP alone (GFP), a GFP fusion with wild-type (WT) ILK, or GFP fusions with L53M, R211C, and R317Q ILK. *ILK* KD-induced cell spreading and focal adhesion defects are reverted by expression of WT GFP-ILK but none of the three mutant constructs. Scale bar, 20 μm. **c** Immunoprecipitation (IP) and Western blot analysis of wild-type (WT) or mutant GFP-ILK and associated PINCH and α-parvin proteins. White arrows indicate GFP-ILK and black arrows endogenous ILK. **d** Confocal images of p-AKT (Ser473) (green) and isolectin B4 (white) staining in control and *Ilk*[iECKO] P6 retina sections. Presence (control) or absence (*Ilk*[iECKO]) of p-AKT in endothelial nuclei (red arrows) in images with or without DAPI signal (blue). Scale bar, 10 μm. **e** qPCR analysis of *AXIN2* expression in HUVEC after siRNA knockdown of endogenous ILK and expression of siRNA-resistant wild-type (WT) or variant ILK proteins, as indicated. Error bars, s.e.m. *p* values (***p* < 0.01, **p* < 0.05, ns not significant), Student's *t* test (*n* = 6). Source data are provided as a Source Data file

Parva mutants[24]. While Itgb1[iECKO] retinal vessels fail to extend endothelial sprouts, mutant ECs do not accumulate in distal focal clusters as those seen in Ilk[iECKO] or Parva[iECKO] retinas. These differences might indicate that other signals, such as growth factor receptors[54], are acting upstream of ILK in ECs. Alternatively, it is also feasible that integrin heterodimers involving other β subunits are responsible for the phenotypic differences. In fact, integrin β3 and β5 suppress pathological angiogenesis even though they are not required for developmental vessel growth[55,56].

Most of the gene mutations observed in FEVR patients affect the Wnt signaling pathway and the corresponding mutant mouse models show vascular defects[1,3,8–11,57] that are strikingly similar to the Ilk[iECKO] phenotype. Indeed, Wnt and ILK cooperate in processes such as hair follicle polarization, skin homeostasis and bone formation[58–60]. ILK was also shown to regulate the stabilization and nuclear translocation of β-catenin, an essential component of the Wnt pathway, in cultured cells[61–63]. Our data show that ILK depletion in ECs reduces β-catenin levels in vitro and in vivo, blunts the cellular response to recombinant hNorrin, and leads to alterations in LEF1 expression in the retinal vasculature. Thus, the two pathways appear to be functional linked in the CNS endothelium, explaining the striking similarities in mutant phenotypes.

While autosomal dominant and X-linked inheritance is most common for FEVR, autosomal recessive inheritance has been also described[34]. The three variants of human ILK identified in patients suffering from exudative vitreoretinopathy represent partial loss-of-function alleles, which raises the question whether development of the disease requires the loss of heterozygosity in ECs. A similar situation has been reported for cerebral cavernous malformation (CCM), a human disease characterized by the development of hyperpermeable vascular malformations in the brain. While familial CCM is caused by heterozygous mutations in one of the three relevant genes, namely CCM1/KRIT1, CCM2, or CCM3/PDCD10, vascular lesions in mice emerge only in homozygous mutants[64,65]. The essential roles of ILK in many cell types and organs might prevent the widespread emergence of homozygous loss-of-function cells early in development, but future work will have to address whether disease onset requires the loss of heterozygosity in ECs or their progenitors. As mutations in the LAMA4 gene, which encodes the extracellular matrix protein laminin-α4, and in ILK (p.Ala262Val; c.C785T) have been associated with human cardiomyopathy[66,67], ILK and other proteins mediating matricellular interactions emerge as key regulators of cardiovascular morphogenesis and might be functionally relevant in FEVR and other human diseases involving the vascular system.

## Methods

**Mice.** Animal experiments were performed in compliance with the relevant laws and institutional guidelines, were approved by local animal ethics committees and were conducted with permissions granted by the Landesamt für Natur, Umwelt und Verbraucherschutz (LANUV) of North Rhine-Westphalia or Animal Experiments of the Ludwig-Maximilians University Munich.

Pdgfb-iCre transgenic mice[23] were bred into a background of Ilk[68] conditional (loxP-flanked) mice, whereas conditional Parva[24] mutants were combined with Cdh5-CreERT2 animals. To induce Cre activity, newborn offspring were given intraperitoneal tamoxifen injections of 50 μg (P1–3) or 100 μg (P3–7) and the phenotypes analyzed at P6 or P14, respectively. Parva[iECKO] mutants were analyzed at P16 after tamoxifen injection at P1–3. In all genetic experiments, tamoxifen-injected littermate animals were used as control, and both male and female animals were used.

For the generation of constitutive Ilk[+/−] heterozygous mice, PGK-Cre transgenic mice were interbred with Ilk[68] conditional (loxP-flanked) animals. Analysis of the resulting mice was done at P6 and P14.

**Immunohistochemistry and EdU labeling of whole-mount retina.** All immunostainings of retina were carried out with littermate processed simultaneously under the same conditions, as described previously[69]. To analyze (and quantitate) the morphology of the retinal vasculature and preserve intact endothelial filopodia and sprouts at the angiogenic front, whole animal eyes were fixed in 4% PFA at 4 °C overnight. The following day, eyes were washed in phosphate-buffered saline (PBS) before retinas were dissected and partially cut in four quadrants. After blocking/permeabilization in 1% bovine serum albumin (BSA) with 0.3% Triton X-100 for several hours at RT or overnight at 4 °C, the retinas were washed two times in Pblec buffer (1% Triton X-100, 1 mM CaCl₂, 1 mM MgCl₂, and 0.1 mM MnCl₂ in PBS [pH 6.8]) for 20 min and incubated for overnight in Pblec containing biotinylated isolectin B4 (VectorLabs, B-1205, 1:50). Following five washes (each 20 min) in blocking solution, retinas were incubated with Alexa Fluor streptavidin-conjugated antibodies (Life Technologies, S11223, 1:100) for 2 h, washed three times more and flat-mounted in microscope glass slides with Fluoromount-G (SouthernBiotech, 0100-01). Double or triple whole-mount immunohistochemistry was performed in retinas fixed 2 h on ice in 4% PFA or in MeOH at −20 °C. After fixation, retinas were blocked for 2 h in 1% BSA with 0.3% Triton X-100 and incubated overnight or for 2 h with isolectin B4 (Vector Labs, B-1205, 1:50) and the following primary antibodies: rat anti-VE-cadherin (BD Biosciences, 555289, 1:200), rabbit anti-Collagen IV (Serotec, 2150-1470, 1:200), rat anti-ICAM2 (BD Biosciences, 553326, 1:200), rabbit anti-Claudin5 (Life Technologies, 341600, 1:200), rat anti-Plvap (Developmental Studies Hybridoma Bank, AB 531797, 1:15), rabbit anti ERG (Abcam, ab110639, 1:100), and LEF1 (Cell signaling, #2230, 1:100). For detection, suitable species-specific Alexa Fluor-coupled secondary antibodies (Invitrogen, 1:500) were used.

**Immunofluorescence staining of retina sections.** Whole retinas were fixed in 4% PFA overnight and dehydrated by stepwise buffer changes to 10, 20, and 30% sucrose in PBS every day. After embedding in Tissue Freezing Medium (Leica), 50 μm sections are obtained using a cryotome (Leica CM3050S). To remove the Tissue Freezing Medium, retina sections were washed twice with PBS at room temperature (RT) for 15 min each. Following overnight blocking in 5% goat serum with 0.3% Triton X-100 in PBS at 4 °C, primary antibodies or lectins were applied in blocking solution: isolectin B4 (Vector Labs, B-1205, 1:50), ILK (Genetex, GTX62096, 1:100), phospho-AKT (Cell signaling, #4060, 1:100) and β-catenin (Cell signaling, #9562, 1:100). For signal detection, suitable species-specific Alexa Fluor-coupled secondary antibodies (Alexa, 1:500) were used with 1:1000 DAPI.

**Image acquisition and processing.** Stained and flat mounted retinas were analyzed at high resolution with Leica TCS SP5 and Zeiss LSM 880 confocal microscopes. A Leica Stereomicroscope MZ16F coupled to a digital camera (Hamamatsu C4742-95) was used for lower resolution. Volocity, IMARIS, Photoshop CS and Illustrator CS (Adobe) software were used for image acquisition and processing. All images shown are representative of at least six different images from three different retina samples per group with identical laser excitation and confocal scanner detection settings.

**NIH3T3 cell culture and colony forming assay.** For the colony formation assay, 6 × 10⁵ of NIH 3T3 cells (Sigma #93061524) were grown in 24-well plates and transfected with 0.8 μg of pHygEF2-based expression plasmid that also contained a hygromycin-resistant marker, using lipofectamine 2000 reagent (Invitrogen). Twenty-four hours after transfection, the cells were trypsinized and seeded into 60-mm dishes. The cells were further grown in D-MEM containing 3% FBS and 100 μg/mL hygromycin for 3 weeks and then stained with crystal violet.

**Cloning of ILK expression vector constructs.** Cloning of ILK expression constructs was performed in the Gateway cloning system (Invitrogen). cDNAs were modified with a CACC sequence in front of the start codon and with three stop codons at the 3′ end to allow site-directed cloning into the pEntry/D-TOPO vector. Mutations were generated by site-directed mutagenesis. The mutagenesis polymerase chain reaction (PCR) was performed using the proof-reading polymerase Pfu (Promega). All expression constructs were sequence verified prior to use in transfection assays.

To construct the pHygEF2/3xHA-h-ILK (wild-type or mutant) plasmid, a double-stranded oligonucleotide encoding the 3xHA-tag was inserted into the pHygEF2 vector after BssHII-ClaI digestion. Then, the human ILK open reading frame was amplified by PCR from pBABE-puro/EGFP-FLAG-h-ILK plasmid using primers (5′-AAAATCGATACTAGTATGGACGACATTT-3′ and 5′-AAATCTAGACTACTTGTCCTGCATCTTC-3′) and inserted into the ClaI-XbaI sites.

**Immunoprecipitation and Western blot.** Cell extracts were prepared by direct addition of 4× sodium dodecyl sulfate (SDS) sample buffer (200 mm Tris–HCl, (pH 6.8), 8% SDS, 400 mm dithiothreitol, 0.2% bromophenol blue, 40% glycerol) to the cells or by cell lysis in lysis buffer (1% NP-40, 1 % Triton-X, 50 mM Tris (pH 8.0), 150 mM NaCl, protease and phosphatase inhibitor cocktail). For immunoprecipitation, anti-GFP (Abcam, ab6556) or anti-FLAG agarose (MBL, FLA-1) was used. After extensive washing with lysis buffer, the precipitates were dissolved in 2× SDS sample buffer. Samples were loaded on SDS–polyacrylamide gel and electrophoresis and Western blotting was performed. Blots were quantified

with the gel analysis function in the program ImageJ. The following antibodies were used for immunoblotting: anti-PINCH (BD, 612710, 1: 5,000), anti-α-parvin (Abcam, ab11336, 1: 5000), anti-ILK (Abcam, ab49979, 1: 5000), anti-tubulin (Sigma, T3526, 1: 5000), anti-FLAG (Sigma, F7425, 1: 5000), anti-β-catenin (Cell signaling, #8814, 1:11,000), anti-AKT (Cell signaling, #4691, 1:5000), and anti-phospho-AKT(ser473) (Cell signaling, #4060, 1:1000). Uncropped versions of the Western blots shown in Fig. 7a, c and Supplementary Fig. 5B as well as additional replicate blots used for the quantitation shown in Supplementary Fig. 5A, B are provided in Supplementary Figs. 6–8.

**Quantitative analysis of the retinal vasculature**. All quantifications were done on high-resolution confocal images representing a thin z-section of the sample. The number of branchpoints and the area covered by ECs were calculated with the Velocity (Perkin Elmer) software from 24 fields sized $770 \times 770$ μm, 6 retina samples per group. The number of endothelial sprouts and filopodial extensions were quantified at the angiogenic front in 24 fields (sized $385 \times 385$ μm, 6 retinas per group) of control and $ILK^{iECKO}$ retinas. The total number of filopodia was normalized for a standard size (100 μm in length) of vessels at the angiogenic front.

EdU labeled, isolectin B4-positive ECs were counted in 16 fields (sized $385 \times 385$ μm, 4 retinas per group). Vascular progression/outgrowth was measured by defining a straight line from the angiogenic front to the center of the retina for each retina quadrant. A minimum of 16 quadrants belonging to 4 retinas per group were used for quantification.

**EC culture and transfection**. HUVECs purchased from Thermo Fisher (C0035C) were maintained in Medium200 (M200) supplemented with low serum growth supplement (Life Technologies). HUVECs at passage 3–6 were used for experiments. HUVECs seeded on 0.2% Gelatin (Sigma) coated dishes at the day before transfection and transfected with 100 nM of siRNAs for human ILK (5′-GCCU-GUGGCUGGACAACACGGAGAA-3′), by using oligofectamine (Life Technologies) according to the manufacturer's instructions. For transient expression of human wild-type or mutant ILK (p.L53M, p.R317Q, and p.R211C) which is siRNA resistant, HUVECs were transfected with one of these vectors or control (without insert) by using oligofectamine (Life Technologies) according to the manufacturer's instructions. Immunostaining, western, or immunoprecipitation were done 48 h after transfection.

**Lentiviral expression vector cloning and virus production**. The coding region of ILK and its mutants which are modified to have resistant to siRNA of ILK were amplified by PCR and cloned into NcoI and SalI sites of pHAGE2-TetO plasmid. pHAGE2-TetO-mCherry was used for estimating the infection rate and control. All constructs were verified extensively by restriction enzyme digestion and sequencing. To produce the lentivirus, HEK293 cells were transfected with 3 μg of psPAX2 (Addgene), 1.5 μg of pMD2.G (Addgene), and 4.5 μg of lentiviral vector using 27 μl of Fugene 6 (Promega) transfection reagent in 600 μl of Opti-MEM. Virus-containing supernatants were collected at 48 h, filtered through a 0.4-μm PVDF filter, concentrated, and resuspended in knockout DMEM and stored at −80 °C until use.

**Norrin**. For stimulation with recombinant Norrin, wild-type, and mutants ILK expression lentivirus infected HUVECs were used which ILK expression is controlled by doxycycline treatment. After lentiviral infection, infected HUVECs were reseeded on 0.2% gelatin coated 60 mm dish. At following day, infected HUVECs were transfected with ILK siRNA by using RNAiMAX (Thermo) manufacturer's instructions for 4 h and change with complete medium with 5 ng/ml of doxycyclin. The knockdown efficiency sustained for 72 h after transfection. Totally, 24 h after siRNA transfection, stimulation with 1.25 μg/ml recombinant human Norrin (R&D Systems) was performed for 8 h and RNA is isolated by RNA is isolated by RNeasy mini kit (Quiagen). qPCR is performed with SsoAdvanced Universal Probes Supermix (BIO-RAD) with Taqman primer: ILK (Hs00177914_m1), Axin2 (Hs00610344_m1), Sox17 (Hs00751752_s1), and FLT1 (Hs01052936_m1).

HRMVECs (ACBRI181, Cell Systems) were transfected with 20 nM of pooled or single siRNA (Dharmacon) with RNAimax. After 24 h, cells were split at 50 K into 6-well plates. Twelve hours later cells were stimulated with 1.25 μg/ml hNorrin for 48 h. Total RNA was extracted using Trizol (Invitrogen) and digested with DNase (Stratagene) and used for RT-qPCR.

**Analysis of phospho-AKT and β-catenin in HUVEC**. Lentivirus infected HUVECs were used for western blotting for analysis of phospho-AKT and β-catenin. At 24 h after seeding of cells in 0.2% gelatin coated 100 mm dishes, lysates were prepared with RIPA buffer containing 1× complete protease inhibitor (Roche) and 1× PhosSTOP (Roche) and quantified by the Bradford assay (Bio-Rad). Western blotting was performed as described above.

**Immunofluorescence analysis of HUVECs**. HUVECs were transferred from a 100 mm dish into a 6-well plate after transfection or infection. Cells were fixed with 4% PFA in RT for 10 min. After 2 rounds of brief washing with PBS, permeabilization/blocking solution (1% BSA with 0.2% triton x-100 in PBS) was added to

the fixed cells for 30 min at RT. Rabbit anti-Paxillin (Cell Signaling, 1:100) and mouse anti-GFP (Chemicon, 1:100) in PBS were administered for 1 h at RT. Following three washes with PBS (5 min each), suitable species-specific Alexa Fluor-coupled secondary antibodies (1:500) were administered togehter with 1:1000 DAPI and samples were incubated for 25 min at RT. After three washes with PBS (5 min each), stained cells were mounted on microscope glass slides with Fluoromount-G (SouthernBiotech, 0100-01).

For labeling of proliferating cells, 100 μg of EdU (Life Technologies) per pup was injected intraperitoneally 4 h before sacrifice. EdU-positive cells were detected with the Click-iT EdU Alexa Fluor-647 Imaging Kit (Life Technologies, C10340).

**DNA sequencing of patients and controls**. Informed consent according to the declaration of Helsinki was obtained from all subjects (208 patients and 258 controls) involved in genetic testing. The human ILK gene consists of 13 exons, 12 of which code for a protein of 452 amino acid residues. Protein coding exons and flanking intronic regions (including splice sites) were amplified by PCR and sequenced using the Sanger chain termination sequencing method. The patients were clinically diagnosed with exudative vitreoretinopathy or a similar disorder. Most of them were sporadic cases and no additional family members were available for segregation analysis of DNA sequence variations.

The study was conducted in accordance with the Helsinki Declaration. The approval for genetic testing was awarded to the Institute of Medical Molecular Genetics by the Federal Office of Public Health (FOPH) in Switzerland. Patients provided informed consent for genetic testing.

**Reporting summary**. Further information on research design is available in the Nature Research Reporting Summary linked to this article.

## Data availability
The source data underlying Figs. 1E, G, 2E, 3A, 4C, 5D, 7E, and Supplementary Figs. 4B, D and 5A–D are provided as a Source Data file. The data that support the findings of this study are available from the corresponding author upon reasonable request.

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

## Acknowledgements

Funding was provided by the Max–Planck-Society, the University of Münster and the German Research Foundation (SFB 1009, SFB 1366, cluster of excellence "Cells in Motion" and project number 391580220). E.M. and A.F. were also supported by the German Research Foundation (MO2562/1–2) and H.J.J. by the NIH (R01 EY024261).

## Author contributions

H.Y., H.P., W.B., and R.H.A. designed the experiments, interpreted results, and wrote the paper. H.Y., H.P., H.J.J., K.-P.K., and K.K. performed the in vitro experiments, L.M., L.A., S.F., and W.B. characterized human exudative vitreoretinopathy patient gDNA samples and constructed mutant cDNAs. H.Y., H.P., I.S., A.F., and E.M. characterized the mutant mice and quantitated results.

## Competing interests

The authors declare no competing interests.
