## [Peer Review File · Nature Communications]

Reviewers' comments:

Reviewer #1 - expert in ILK (Remarks to the Author):

In this paper, Yamamoto et al demonstrate that Integrin-Linked Kinase (ILK) is essential for retinal angiogenesis. Inducible knock-out of the ILK gene in post-natal endothelial cells results in defects in retinal angiogenesis resembling Familial exudative vitreoretinopathy (FEVR), a human disease characterized by defective retinal angiogenesis and associated complications that can result in complete vision loss.

The authors have also screened genomic DNA samples from FEVR patients for mutations in human ILK, and have identified 3 distinct mutations within all three domains of the ILK protein: ankyrin repeat, PH domain, and kinase domain.

While the role of ILK in embryonal vascularization and tumour angiogenesis has been demonstrated previously, this work is quite novel in demonstrating an essential role of ILK in post-natal vascularization and specifically in retinal angiogenesis and the pathology of FEVR.

Overall the work is of sound quality, and most of the data are of high quality and well presented.

However, from the mechanistic perspective, the paper could be improved by addressing the following :

1. The authors refer extensively to the role of Wnt signaling in FEVR. ILK has been shown in numerous studies to influence the Wnt signaling pathway, specifically beta-catenin stabilization, nuclear localization and beta-catenin mediated transcription. Given the overlap between the Wnt FEVR and ILK FEVR phenotypes, the authors should examine the status of beta-catenin and TCF activity in the mouse models and in the ILK mutant transfected HUVECs.
2. In Fig 1G, and in Fig 7D there is a clear effect of ILK KO or mutations in EC and NIH3T3 proliferation. Given that ILK is an established regulator of cell cycle and cyclin D1 expression, the status of cell cycle, cyclins and cdk activities would add considerably to the mechanistic defects of ILK KO and mutations.
3. The authors point out that R211 within the PH domain of ILK has been shown to play a critical role in ILK mediated activation of Akt (through regulation of Akt Ser 473 phosphorylation) (ref 40). Since Akt signaling is critical for the regulation of cell survival and migration, the authors should examine the status of Akt phosphorylation in the ILK KO retina and EC cells as well as in ILK mutant transfected HUVECs
4. Recently a new interactor of ILK, called LIMD2, has been described: LIMD2 binds directly to ILK within the kinase domain and this interaction is essential for the pro-migratory role of ILK.

Furthermore, in vitro, LIMD2 stimulates ILK activity (Peng H et al, Cancer Res. 74, 1390, 2014). The authors should determine whether the R317Q ILK mutant is defective in LIMD2 binding (by co-immunoprecipitation). If this were the case, this would provide additional , critical mechanistic data for the requirement of ILK in post-natal vascularization and related pathologies such as FEVR

Minor points:

1. On page 4, the authors state: " However, the role of ILK in the postnatal endothelium has not been studied so far and therefore its functional role in angiogenesis in vivo remains little understood" The authors should qualify this statement by citing work such as: Tan,C et al Cancer Cell, 5, 79, 2004, in which a role of ILK in tumour angiogenesis has been established.
2. The y-axis of the bar graph in Fig 7D needs description/labeling

Reviewer #2 - expert in vitreoretinopathy (Remarks to the Author):

Comments to manuscript 'Integrin-linked kinase deficiency is a cause of exudative vitreoretinopathy' by Yamamoto et al.

In this manuscript, the authors describe defects in retinal vascularisation within a conditional knock-out mouse model for integrin-linked kinase (Ilk), and claim the identification of three mutations in human ILK that would underlie the condition familial exudative vitreoretinopathy (FEVR). Whereas the findings are novel, and the experiments in general have been well-conducted and well-described, I am missing a number of essential experiments that link the phenotype of the IlkiECKO mice to the absence / reduced expression of Ilk. In addition, I am not fully convinced by the causative nature of the three ILK variants described to underlie FEVR. Below are my specific comments.

Major points:

1. Title: as I am not convinced by the causative nature of the three described missense variants in human ILK (point 8), I also do not agree to the Title.

2. Abstract, page 2, line 44-46: I do not agree to this sentence, as especially the work done on the Norrin/beta-catenin signaling pathway (NDP, FZD4, LRP5 and TSPAN12) has largely increased our insights on the pathophysiological mechanisms underlying FEVR. Please rephrase.

3. One of my major concerns involves the lack of data showing that the addition of tamoxifen in the IlkiECKO mouse model indeed results in a decreased expression or function of Ilk. A Q-PCR, Western blot and/or immunostaining for ILK showing that indeed Ilk transcripts or proteins are reduced would be essential to link the bona fide phenotype in the mice directly to Ilk.

4. Figures 1, 3 and Supplemental Figures 1 and 3 show quite some immunostainings, although from the description, it is sometimes not clear what exactly is stained and, more importantly, what it implicates. A nice example of how it should be explained is on page 7, lines 160-161, where the authors state that "Expression of Pvlap/PV1, which forms diaphragms in endothelial fenestrae....". Please use similar descriptions for all markers used.

5. Page 5, line 127. All of a sudden, the authors switch from a single tamoxifen delivery to three consecutive administrations. Please explain the rationale for this.

6. Page 7, line 176. The use of the ParvaiECKO model comes out of the blue. Please spend one or two sentences on why this model was selected.

7. Page 7, line 182 and Discussion. Given the importance of heterozygous changes detected in humans, and the presence of a somewhat milder vascular phenotype in heterozygous IlkiECKO/+ mice, I'm totally lacking a discussion of what is observed in constitutive Ilk+/- mice.

8. My other major concern involves the relatively high frequency for the ILK variants described in Exac, especially for p.Arg211Cys and also for p.Leu53Met. Although the authors claim that the occurrence of these variants are not extremely frequent, I think this frequency is too high to underlie a dominant condition such as FEVR. For instance, for p.Arg211Cys, with a frequency of 0.084%, 8 out of 10.000 (appx. 1:1,250 individuals) would have FEVR, only due to this mutation (assuming full penetrance). This is way too high for a rare dominant condition such as FEVR. In addition, the authors do not demonstrate nor discuss any segregation analysis, presence of this variant within healthy or affected relatives, or whether these variants occur de novo or are inherited. Together, despite the fact that the authors clearly demonstrate that the missense variants can affect ILK protein function (Figure 7), the conclusion that these variant underlie FEVR to me is not sufficiently proven.

9. The Discussion section is short, and lacks certain components, some of which are addressed above. Also, the discussion on the *Itgb1iECKO* model now appears unexpected. Finally, the last paragraph of the Discussion (linking ILK to cardiomyopathy) to me feels inappropriate at the very end of a paper on vitreoretinopathy.

10. Figures 1a and 2a: I propose to expand these figures, by not only putting the tamoxifen but also showing the schematic build-up of the *IlkiECKO* model (especially in 1a), and indicate an arrow for each tamoxifen delivery (in figure 2a).

Minor issues:

11. Page 5, line 116: the sentence starting with confirming should not be a separate sentence but rather an extension of the previous one.

12. The positioning of Figure 5 feels a bit inappropriate, I would propose to combine Figures 2 and 5 into a single Figure 2.

13. Supplemental Figure 3 is not at all mentioned in the main text.

14. The yellow traces in Figure 6a are hard to see, please use a different coloring.

Reviewer #3 - expert in retinal angiogenesis (Remarks to the Author):

This paper by Yamamoto et al. investigates the role of integrin-linked kinase (Ilk) in familial exudative vitreoretinopathy (FEVR). Inducible endothelial cell (EC)-specific *Ilk* KO mice were generated and analyzed phenotypically, which demonstrated defective retinal vasculature resembling FEVR symptoms. Genetics analysis in FEVR patients identified three loss-of-function mutations in *Ilk*, which showed impaired biological function in protein binding and vascular endothelial cell spreading and

focal adhesion. This is a well conducted and convincing study identifying Ilk as a new disease gene for FEVR, which is linked largely with Wnt signaling defects previously. The novel findings are of high interest to researchers in the fields of vascular biology, ophthalmology and eye research. There are just a few concerns from this reviewer, and addressing them may help further strengthen the manuscript.

Major concerns:

1. Where is endogenous Ilk localized in the retina? The inducible Ilk KO data presumes vascular endothelial localization but that assumption is not proven with immunohistochemistry. Can Ilk be expressed elsewhere in the retina to exert vascular effects?
2. The EC specificity of the PDGFb-iCre mice needs to be established, eg. by crossing with a reporter strain to exclude the potential function of Ilk in other retinal cell types (glia and neurons) in causing the vascular abnormalities. Validation of successful Ilk knockout in retinal ECs will also be helpful.
3. Another primary feature of FEVR is persistent hyaloid. Do the Ilk iECKO eyes show delayed hyaloid regression?
4. Mechanistically, it will be valuable to delineate whether Ilk signaling is somehow linked with Wnt signaling and actin dynamics. Previously mice with defective Wnt signaling and Akt/Girdin (an actin-binding protein) signaling also showed very similar impairment in retinal vasculature. Do Ilk knockout retinas/ECs have impaired Wnt signaling or actin cytoskeleton dynamics, eg. altered phosphorylation of b-catenin and any changes in actin/Girdin?

Minor concerns:

1. Fig. 5. The vascular phenotype in heterozygous mice is very mild. Do they have reduced body weight, and if so, how can the authors rule out delayed development as a possible cause of secondary vascular developmental delay in hets? It is of interest to note that most other heterozygous Wnt deficient FEVR mice (Fzd4^{+/-} and Lrp5^{+/-}) are phenotypically normal. Please comment.
2. Fig. 7D, define M1, M2, M3.
3. Define abbreviation the first time of use in the main texts (FEVR, line 61); Line 190, FEVR patients (missing an "F").

We would like to thank all reviewers for their time, effort and valuable suggestions, which are greatly appreciated and have enabled us to improve the manuscript further. While a detailed point-by-point response to all comments and questions is provided below, we would like to emphasize that a substantial amount of new results has been added to the manuscript. Most importantly, we have added new data showing that the loss of ILK leads to alterations in Wnt signaling.

We would also like to apologize for the overly long time taken up by the revision. This was caused by breeding issues in our *Ilk* mouse colony, which have substantially delayed all *in vivo* experiments.

Reviewer #1

In this paper, Yamamoto et al demonstrate that Integrin-Linked Kinase (ILK) is essential for retinal angiogenesis. Inducible knock-out of the ILK gene in post-natal endothelial cells results in defects in retinal angiogenesis resembling Familial exudative vitreoretinopathy (FEVR), a human disease characterized by defective retinal angiogenesis and associated complications that can result in complete vision loss.

The authors have also screened genomic DNA samples from FEVR patients for mutations in human ILK, and have identified 3 distinct mutations within all three domains of the ILK protein: ankyrin repeat, PH domain, and kinase domain.

While the role of ILK in embryonal vascularization and tumour angiogenesis has been demonstrated previously, this work is quite novel in demonstrating an essential role of ILK in post-natal vascularization and specifically in retinal angiogenesis and the pathology of FEVR.

Overall the work is of sound quality, and most of the data are of high quality and well presented.

However, from the mechanistic perspective, the paper could be improved by addressing the following :

Question 1. *The authors refer extensively to the role of Wnt signaling in FEVR.*

ILK has been shown in numerous studies to influence the Wnt signaling pathway, specifically beta-catenin stabilization, nuclear localization and beta-catenin mediated transcription. Given the overlap between the Wnt FEVR and ILK FEVR phenotypes, the authors should examine the status of beta-catenin and TCF activity in the mouse models and in the ILK mutant transfected HUVECs.

Reply: We agree with the reviewer that this is an important question. In the revised manuscript, we show strongly reduced endothelial beta-catenin immunosignals in *Ilk*^{IECKO} retina relative to littermate control sections (Fig. 3E). Strong expression and nuclear localization of LEF1, a member of the TCF/LEF transcription factor family that binds to beta-catenin and mediates canonical Wnt signaling, marks control but not *Ilk*^{IECKO} mutants veins (Suppl. Fig. 2C). PVLAP expression, which is normally suppressed by beta-catenin, is increased in *Ilk*^{IECKO} retinal vessel. *In vitro*, we found that Wnt target genes are downregulated by after siRNA-mediated *ILK* knockdown both in human umbilical vein endothelial cells (HUVECs) and HRECs (human retinal ECs) (Fig. 7E and Suppl. Fig. 5B). *ILK* knockdown in HUVECs also results in a reduction of beta-catenin levels (Fig. 7A), which is rescued by the re-expression of siRNA-resistant WT *ILK* but not by any of the 3 *ILK* variant proteins. Together, these data indicate that *ILK* is required for canonical Wnt signaling, which provides a mechanistic explanation for the striking phenotypic similarities between *Ilk* and Wnt pathway loss-of-function mutants.

Question 2. *In Fig 1G, and in Fig 7D there is a clear effect of ILK KO or mutations in EC and NIH33 proliferation. Given that ILK is an established regulator of cell cycle and cyclin D1 expression, the status of cell cycle, cyclins and cdk activities would add considerably to the mechanistic defects of ILK KO and mutations.*

Reply: We thank the reviewer for this question. EC-specific knockout of *ILK* reduces endothelial cell proliferation in the postnatal retina (Fig. 1F, G). To address whether the cell cycle is altered after depletion of *ILK* in HUVECs, we also did a cell cycle analysis using propidium iodide staining. However, in contrast to what has been previously found in epithelial cells (D'Amico et al. 2000, J Biol Chem. 275:32649-5), the cell cycle in HUVECs is not substantially altered by the knockdown of *ILK* (see below, left panel). Likewise, *ILK* siRNA treatment does not

alter *CCND1* transcript levels in HUVECs. This finding is consistent with a previous report showing that Wnt3a-dependent beta-catenin activation does not increase *CCND1* expression in HUVECs (Samarzija et al. 2009, *Biochem Biophys Res Commun.* 386:449-54). Thus, we cannot conclude that ILK plays a direct role in cell cycle regulation in endothelial cells.

Question 3. The authors point out that R211 within the PH domain of ILK has been shown to play a critical role in ILK mediated activation of Akt (through regulation of Akt Ser 473 phosphorylation) (ref 40). Since Akt signaling is critical for the regulation of cell survival and migration, the authors should examine the status of Akt phosphorylation in the ILK KO retina and EC cells as well as in ILK mutant transfected HUVECs.

Reply: We found a downregulation of phospho-AKT immunosignals in *Ilk*^{IECKO} retinal endothelial cells relative to littermate control (new data in Fig. 7D). In HUVECs, *ILK* siRNA treatment reduces Akt Ser473 phosphorylation, which is rescued by re-expression of siRNA-resistant WT ILK but not any of the point mutants (Fig. 7A).

Question 4. Recently a new interactor of ILK, called LIMD2, has been described: LIMD2 binds directly to ILK within the kinase domain and this interaction is essential for the pro-migratory role of ILK. Furthermore, in vitro, LIMD2 stimulates ILK activity (Peng H et al, *Cancer Res.* 74, 1390, 2014). The authors should determine whether the R317Q ILK mutant is defective in LIMD2 binding (by co-immunoprecipitation). If this were the case, this would provide additional, critical

mechanistic data for the requirement of ILK in post-natal vascularization and related pathologies such as FEVR.

Reply: It will be indeed interesting to study the role of LIMD2 or other interaction partners of ILK in retinal angiogenesis, but, in our view, these questions deserve a separate investigation and are beyond the scope of the current manuscript.

Question 5. On page 4, the authors state: " However, the role of ILK in the postnatal endothelium has not been studied so far and therefore its functional role in angiogenesis in vivo remains little understood" The authors should qualify this statement by citing work such as: Tan,C et al Cancer Cell, 5, 79, 2004, in which a role of ILK in tumour angiogenesis has been established.

Reply: We thank the reviewer for this suggestion and have incorporated the reference into the introduction (see page 4) of the revised manuscript (ref. 21).

Question 6. The y-axis of the bar graph in Fig 7D needs description/labeling

Reply: This has been corrected. The data is now shown as Suppl. Fig. 5C.

Reviewer #2

Comments to manuscript 'Integrin-linked kinase deficiency is a cause of exudative vitreoretinopathy' by Yamamoto et al.

In this manuscript, the authors describe defects in retinal vascularisation within a conditional knock-out mouse model for integrin-linked kinase (Ilk), and claim the identification of three mutations in human ILK that would underlie the condition familial exudative vitreoretinopathy (FEVR). Whereas the findings are novel, and the experiments in general have been well-conducted and well-described, I am

missing a number of essential experiments that link the phenotype of the IlkiECKO mice to the absence / reduced expression of Ilk. In addition, I am not fully convinced by the causative nature of the three ILK variants described to underlie FEVR. Below are my specific comments.

Major points:

Question 1. *Title: as I am not convinced by the causative nature of the three described missense variants in human ILK (point 8), I also do not agree to the Title.*

Reply: We agree that it is difficult to prove that any of the 3 missense variants is indeed causing exudative vitreoretinopathy. Human pedigree analysis would be helpful in this context, but, as we point out in the manuscript, the mutations were found in spontaneous cases. Despite of these limitations, our data – namely the *Ilk* loss-of-function phenotype, the similarities to Wnt pathway mutants, the link between ILK and Norrin-induced cellular responses – argue for a role of ILK in exudative vitreoretinopathy.

To address the reviewer's concern, we have modified some parts of the manuscript and changed the title, which now reads "Integrin-linked kinase controls retinal angiogenesis and is linked to exudative vitreoretinopathy".

Question 2. *Abstract, page 2, line 44-46: I do not agree to this sentence, as especially the work done on the Norrin/beta-catenin signaling pathway (NDP, FZD4, LRP5 and TSPAN12) has largely increased our insights on the pathophysiological mechanisms underlying FEVR. Please rephrase.*

Reply: We agree and did not mean to question that there is a substantial body of research on the important role of Norrin/beta-catenin signaling in the pathophysiological mechanisms underlying FEVR. We have modified the abstract and introduction to avoid any misunderstandings.

Question 3. *One of my major concerns involves the lack of data showing that the addition of tamoxifen in the Ilk^{iECKO} mouse model indeed results in a decreased expression or function of Ilk. A Q-PCR, Western blot and/or immunostaining for ILK showing that indeed Ilk transcripts or proteins are reduced would be essential to link the bona fide phenotype in the mice directly to Ilk.*

Reply: Agree. We have added new data showing ILK immunostaining in P6 retina after three tamoxifen injections at postnatal day (P) 1-3 (see scheme in Fig. 1A).

While ILK expression in perivascular cells is maintained in *Ilk^{iECKO}* retinal samples, endothelial signal is lost (Suppl. Fig. 1A).

Question 4. *Figures 1, 3 and Supplemental Figures 1 and 3 show quite some immunostainings, although from the description, it is sometimes not clear what exactly is stained and, more importantly, what it implicates. A nice example of how it should be explained is on page 7, lines 160-161, where the authors state that “Expression of Pvlap/PV1, which forms diaphragms in endothelial fenestrae....”.*

Please use similar descriptions for all markers used.

Reply: We are thankful for this comment and have revised the manuscript accordingly.

Question 5. *Page 5, line 127. All of a sudden, the authors switch from a single tamoxifen delivery to three consecutive administrations. Please explain the rationale for this.*

Reply: In both treatment regimes, which are now shown in diagrams (See Fig. 1A and Fig. 2A), three consecutive tamoxifen injections were used. As mentioned in the manuscript, we switched to the second scheme (tamoxifen at P3, 5 and 7 followed by analysis at P14) to analyze the vascularization of the deeper retina

and also circumvent the limited survival observed after early postnatal tamoxifen administration. This is an important experimental refinement limiting adverse effects in the mutant animals. At the same time, we make sure that we are not analyzing indirect, secondary defects caused by poor health of the mutant animals.

Question 6. *Page 7, line 176. The use of the ParvaiECKO model comes out of the blue. Please spend one or two sentences on why this model was selected.*

Reply: ILK, PINCH and parvin form heterotrimeric IPP complexes, which also lead to the stabilization of the three proteins each other (see text on top of page 9). Previous work has linked alpha-parvin (encoded by the *Parva* gene) to angiogenesis and the stabilization of endothelial junctions. Thus, it makes a lot of sense to compare the defects seen in EC-specific *Ilk* and *Parva* mutants similar to what has been done for different Wnt pathway components (e.g. Wang et al. 2012, Cell 151:1332-1344). In the text, we explain the relevance of parvin as a component of IPP complexes and also mention that *Parva* has been previously linked to angiogenesis.

Question 7. *Page 7, line 182 and Discussion. Given the importance of heterozygous changes detected in humans, and the presence of a somewhat milder vascular phenotype in heterozygous *IlkiECKO/+* mice, I'm totally lacking a discussion of what is observed in constitutive *Ilk+/-* mice.*

Reply: The phenotype of *Ilk^{iECKO/+}* heterozygotes suggests some degree of haploinsufficiency in retinal vascularization. With regard to constitutive (global) mutants, it has been reported that these heterozygotes do not show defects during embryonic development (Sakai et al. 2003, Genes Dev 17:926-940) and are viable and fertile, whereas postnatal vascular development in the retina has not been analyzed so far. The revised manuscript contains new data showing that constitutive loss of a single *Ilk* allele leads to similar defects in the P14 retina (Suppl. Fig. 4A-D) as what we report for EC-specific *Ilk^{iECKO/+}* heterozygotes. These results further support the notion that endothelial cells respond very

sensitively to changes in ILK function.

Question 8. My other major concern involves the relatively high frequency for the ILK variants described in Exac, especially for p.Arg211Cys and also for p.Leu53Met. Although the authors claim that the occurrence of these variants are not extremely frequent, I think this frequency is too high to underlie a dominant condition such as FEVR. For instance, for p.Arg211Cys, with a frequency of 0.084%, 8 out of 10,000 (appx. 1:1,250 individuals) would have FEVR, only due to this mutation (assuming full penetrance). This is way too high for a rare dominant condition such as FEVR. In addition, the authors do not demonstrate nor discuss any segregation analysis, presence of this variant within healthy or affected relatives, or whether these variants occur de novo or are inherited. Together, despite the fact that the authors clearly demonstrate that the missense variants can affect ILK protein function (Figure 7), the conclusion that these variant underlie FEVR to me is not sufficiently proven.

Reply: First of all, we would like to emphasize that, depending on the gene in question, FEVR is can be inherited as autosomal dominant, X-linked or autosomal recessive. Thus, even though X-linked or autosomal dominant inheritance is frequent, this is not a mandatory feature of the disease. In fact, given the critical roles of ILK in early embryonic development, it would be very surprising to find dominant mutations in human subjects. Furthermore, a fraction of subjects carrying *ILK* mutations may be actually asymptomatic, as is, for example, also the case for cerebral cavernous malformations (e.g. Velz et al. 2018, Front Neurol. 9: 848). Another interesting aspect concerns the population frequency of the mutants. According to ExAC, p.Leu53Met is 10 times more frequent in the Finnish population than among non-Finnish Europeans. The mutation has not been found in African or East Asian subjects. p.Arg317Glu is actually quite rare, while p.Arg211Cys is most frequent in the Latino population.

We do not dispute that further genetic studies are required to establish whether mutations in the human *ILK* gene can cause exudative vitreoretinopathy or function as a modifier in the context of Wnt pathway mutations. As we have no evidence for familial inheritance of ILK mutations in humans, we have revised

the manuscript accordingly.

Question 9. The Discussion section is short, and lacks certain components, some of which are addressed above. Also, the discussion on the *Itgb1iECKO* model now appears unexpected. Finally, the last paragraph of the Discussion (linking ILK to cardiomyopathy) to me feels inappropriate at the very end of a paper on vitreoretinopathy.

Reply: We have revised the Discussion to improve the connection between the different paragraphs. The reason for discussing integrin function in the endothelium is the important connection between integrins and ILK in many different settings and cell types. We also think that it is important to mention other pathophysiological settings related to *ILK* mutations in humans.

Question 10. Figures 1a and 2a: I propose to expand these figures, by not only putting the tamoxifen but also showing the schematic build-up of the *IlkiECKO* model (especially in 1a), and indicate an arrow for each tamoxifen delivery (in figure 2a).

Reply: Agree. Schematic diagrams are now shown in Fig. 1A and Fig. 2A.

Minor issues:

Question 11. Page 5, line 116: the sentence starting with confirming should not be a separate sentence but rather an extension of the previous one.

Reply: We have rephrased this sentence.

Question 12. The positioning of Figure 5 feels a bit inappropriate, I would propose to combine Figures 2 and 5 into a single Figure 2.

Reply: We have considered this suggestion, but, in our view, this is still the right place for this figure (after the EC-specific loss-of-function experiments and before the characterization of the human hypomorphic mutations). Moreover, Figures 2 and 5 are too large and cannot be easily merged with compromising the quality and resolution of the images.

Question 13. Supplemental Figure 3 is not at all mentioned in the main text.

Reply: Thank you for alerting us to this issue, which has been addressed.

Question 14. The yellow traces in Figure 6a are hard to see, please use a different coloring.

Reply: As requested by the reviewer, we have changed the color in Fig. 6A.

Reviewer #3

This paper by Yamamoto et al. investigates the role of integrin-linked kinase (Ilk) in familial exudative vitreoretinopathy (FEVR). Inducible endothelial cell (EC)-specific Ilk KO mice were generated and analyzed phenotypically, which demonstrated defective retinal vasculature resembling FEVR symptoms. Genetics analysis in FEVR patients identified three loss-of-function mutations in Ilk, which showed impaired biological function in protein binding and vascular endothelial cell spreading and focal adhesion. This is a well conducted and convincing study identifying IIK as a new disease gene for FEVR, which is linked largely with Wnt signaling defects previously. The novel findings are of high interest to researchers in the fields of vascular biology, ophthalmology and eye research. There are just a few concerns from this reviewer, and addressing them may help further strengthen the manuscript.

Major concerns:

Question 1. *Where is endogenous Ilk localized in the retina? The inducible Ilk KO data presumes vascular endothelial localization but that assumption is not proven with immunohistochemistry. Can Ilk be expressed elsewhere in the retina to exert vascular effects?*

Reply: ILK is expressed widely expressed by many different cell types. For the retina, we have added new data showing anti-ILK immunosignals in endothelial but also perivascular cells. Endothelial ILK signal is lost in *Ilk*^{IECKO} retina samples, whereas perivascular staining is unaffected (Suppl. Fig. 1A).

Question 2. *The EC specificity of the PDGFb-iCre mice needs to be established, eg. by crossing with a reporter strain to exclude the potential function of Ilk in other retinal cell types (glia and neurons) in causing the vascular abnormalities. Validation of successful ILK knockout in retinal ECs will also be helpful.*

Reply: The Mouse Genome Informatics (MGI) database currently lists 89 publications that have used this line for functional studies in the endothelium. With regard to the EC-specificity in the central nervous system and, in particular, the retina, we would like to refer to our own work (Benedito et al. 2009, Cell 137:1124-35; see data below) but also other numerous other studies in the field (e.g. Wang et al. 2019, Elife 8. pii: e43257).

Figure S2. Pattern of *Pdgfb-iCreER* expression in the retinal vasculature

Confocal images of whole-mount P6 retina stained with anti-isolectin B4 (IsolB4, red) and anti-GFP antibodies (green), detecting the expression of an IRES-GFP in the *Pdgfb-iCreER* transgene (A, C), or just the isolated green channels (B, D) are shown. Prominent signal is visible throughout the retinal vasculature including arteries, vein, capillaries and tip cells.

Nuclei in (C, D) were stained with Topro-3.

Scale bars: A, B: 150 μ m; C, D: 40 μ m.

Question 3. Another primary feature of FEVR is persistent hyaloid. Do the *Ilk*^{iECKO} eyes show delayed hyaloid regression?

Reply: As in many other mouse models with defective vascularization of the retina, regression of hyaloid vessels is compromised in P14 *Ilk*^{iECKO} mutants (see image for reviewers below). Given that this might reflect a general, secondary defect that may not be directly related to EC-specific *Ilk* inactivation, we have not included this data in the revised manuscript.

Question 4. Mechanistically, it will be valuable to delineate whether *Ilk* signaling is somehow linked with Wnt signaling and actin dynamics. Previously mice with defective Wnt signaling and Akt/Girdin (an actin-binding protein) signaling also showed very similar impairment in retinal vasculature. Do *Ilk* knockout retinas/ECs have impaired Wnt signaling or actin cytoskeleton dynamics, eg. altered phosphorylation of b-catenin and any changes in actin/Girdin?

Reply: We agree with the reviewer and have investigated possible links between ILK and Wnt signaling. In the revised manuscript, we show strongly reduced endothelial beta-catenin immunosignals in *Ilk*^{iECKO} retina relative to littermate control sections (Fig. 3E). Strong expression and nuclear localization of LEF1, a member of the TCF/LEF transcription factor family that binds to beta-catenin and mediates canonical Wnt signaling, marks control but not *Ilk*^{iECKO} mutants veins (Suppl. Fig. 2C). PVLAP expression, which is normally suppressed by beta-catenin, is increased in *Ilk*^{iECKO} retinal vessel. *In vitro*, we found that Wnt target genes are downregulated by after siRNA-mediated *ILK* knockdown both in human umbilical vein endothelial cells (HUVECs) and HRECs (human retinal ECs) (Fig. 7E and Suppl. Fig. 5B). *ILK* knockdown in HUVECs also results in a reduction of beta-catenin levels (Fig. 7A), which is rescued by the re-expression of siRNA-resistant WT *ILK* but not by any of the 3 *ILK* variant proteins. Together, these data indicate that *ILK* is required for canonical Wnt signaling, which provides a

mechanistic explanation for the striking phenotypic similarities between *Ilk* and Wnt pathway loss-of-function mutants.

Girdin is known to regulate endothelial cell migration and angiogenesis in the postnatal retina is compromised in knockout mice (Kitamura et al. 2008, Nat Cell Biol. 10:329-37). Given that Girdin is a substrate of Akt/PKB, it is quite likely that its function is altered in the absence ILK. This interesting question, however, deserves a separate investigation and is beyond the scope of the current manuscript.

Minor concerns:

Question 5. *Fig. 5. The vascular phenotype in heterozygous mice is very mild. Do they have reduced body weight, and if so, how can the authors rule out delayed development as a possible cause of secondary vascular developmental delay in hets? It is of interest to note that most other heterozygous Wnt deficient FEVR mice (*Fzd4*^{+/-} and *Lrp5*^{+/-}) are phenotypically normal. Please comment.*

Reply: The body weight of heterozygous mutants is slightly reduced but this effect is not statistically significant. *Ilk* heterozygotes and littermate controls show a similar size of the eye and normal formation of the superficial vascular plexus, arguing against general developmental delay.

Question 6. *Fig. 7D, define M1, M2, M3.*

Reply: Thank you very much for alerting us to this issue. The labeling of this panel (now Suppl. Fig. 5C) has been revised.

Question 7. *Define abbreviation the first time of use in the main texts (FEVR, line 61); Line 190, FEVR patients (missing an "F").*

Reply: FEVR is introduced in the first line of the abstract. We sometimes refer to exudative vitreoretinopathy (and not FEVR) because there is currently no evidence for familial inheritance of *ILK* mutations.

REVIEWERS' COMMENTS:

Reviewer #1 (Remarks to the Author):

The authors have responded to most of my concerns and have improved the paper significantly. The addition of new data demonstrating that ILK regulates Wnt/beta-catenin signaling, as well as Akt signaling in retinal angiogenesis, now adds mechanistic insights to this important process. The new data are convincing and of good quality.

I propose a couple of minor revisions:

1. In light of the new data linking ILK to Wnt/beta-catenin, the authors should add a sentence in the Abstract to reflect this mechanistic insight, especially due to previous reports implicating this pathway in exudative vitreoretinopathy.
2. The Western blots in Fig 7A and Suppl Fig 5A require quantification
3. I agree with the authors in discussing the role of ILK in other cardiovascular pathologies and suggest that they add the following recent reference for completion : Rezazadeh, BA et al, Mutations in ILK encoding integrin-linked kinase, are associated with arrhythmogenic cardiomyopathy, *Transl. Res.*, PMID: 30802431

Reviewer #2 (Remarks to the Author):

The authors have performed a substantial set of additional experiments, and thereby significantly strengthened the manuscript. Most of my own concerns, as well as those from two other reviewers, have been adequately addressed.

My only remaining concern is the title that, based on my previous comments, has already been modified. However, I propose to slightly amend it further to '....angiogenesis and MAY BE linked to exudative vitreoretinopathy'. In fact, the authors themselves also tone down their conclusion on a direct involvement of the ILK variants in exudative vitreoretinopathy, hence this comment.

Reviewer #3 (Remarks to the Author):

The authors performed additional experiments that strengthened the manuscript. All concerns are addressed.

Once again, we would like to thank all reviewers for their time, effort and valuable suggestions, which are greatly appreciated and have enabled us to improve the manuscript further. Please see our point-by-point response to all remaining comments and questions below:

Reviewer #1

Question 1. *The authors have responded to most of my concerns and have improved the paper significantly. The addition of new data demonstrating that ILK regulates Wnt/beta-catenin signaling, as well as Akt signaling in retinal angiogenesis, now adds mechanistic insights to this important process. The new data are convincing and of good quality.*

I propose a couple of minor revisions:

1. In light of the new data linking ILK to Wnt/beta-catenin, the authors should add a sentence in the Abstract to reflect this mechanistic insight, especially due to previous reports implicating this pathway in exudative vitreoretinopathy.

Reply: We agree but we also feel that this aspect is already sufficiently covered by the last sentence of the abstract. In addition, there is a size limitation for the abstract (150 words) so that we have to keep this part of the manuscript short. To emphasize the link between ILK and Wnt signaling, we have, however, modified the title of the article accordingly.

Question 2. *The Western blots in Fig 7A and Suppl Fig 5A require quantification*

Reply: Quantitation of the data is now provided in Supplementary Fig. 5A and B.

Question 3. *I agree with the authors in discussing the role of ILK in other cardiovascular pathologies and suggest that they add the following recent reference*

for completion : Rezazadeh,BA et al, Mutations in ILK eccoding integrin-linked kinase, asre associated with arrhymogeniv cardiomyopathy, Tranl. Res., PMID: 30802431

Reply: We have included this reference at the end of the Discussion.

Reviewer #2

Question 1. *The authors have performed a substantial set of additional experiments, and thereby significantly strengthened the manuscript. Most of my own concerns, as well as those from two other reviewers, have been adequately addressed.*

My only remaining concern is the title that, based on my previous comments, has already been modified. However, I propose to slightly amend it further to '...angiogenesis and MAY BE linked to exudative vitreoretinopathy'. In fact, the authors themselves also tone down their conclusion on a direct involvement of the ILK variants in exudative vitreoretinopathy, hence this comment.

Reply: We appreciate the reservations of the reviewer, but we also feel that the current phrasing is sufficiently cautious and supported by our data. The work “link” reflects some kind of connection without claiming that there is a causal relationship. The striking phenotypic similarities with established animal models of FEVR and the alterations in Wnt signaling, which is now also mentioned in the title, are, in our view, sufficient basis for this important conceptual claim.

Reviewer #3

The authors performed additional experiments that strengthened the manuscript. All concerns are addressed.

Reply: We thank the reviewer for the kind comment.